# The Effect of Climate Change and the Snail-Schistosome Cycle in Transmission and Bio-Control of Schistosomiasis in Sub-Saharan Africa

**DOI:** 10.3390/ijerph17010181

**Published:** 2019-12-26

**Authors:** Tayo Alex Adekiya, Raphael Taiwo Aruleba, Babatunji Emmanuel Oyinloye, Kazeem Oare Okosun, Abidemi Paul Kappo

**Affiliations:** 1Biotechnology and Structural Biology Group, Department of Biochemistry and Microbiology, University of Zululand, KwaDlangezwa 3886, South Africa; arulebataiwo@yahoo.com (R.T.A.); tunji4reele@yahoo.com (B.E.O.); 2Department of Biochemistry, Afe Babalola University, PMB 5454, Ado-Ekiti 360001, Nigeria; 3Department of Mathematics, Vaal University of Technology, Vanderbijlpark 1900, South Africa; kazeemoare@googlemail.com

**Keywords:** *Biomphalaria* spp., *Bulinus* spp., cercariae, climate change, Schistosomiasis, sub-Saharan Africa

## Abstract

In the next century, global warming, due to changes in climatic factors, is expected to have an enormous influence on the interactions between pathogens and their hosts. Over the years, the rate at which vector-borne diseases and their transmission dynamics modify and develop has been shown to be highly dependent to a certain extent on changes in temperature and geographical distribution. Schistosomiasis has been recognized as a tropical and neglected vector-borne disease whose rate of infection has been predicted to be elevated worldwide, especially in sub-Saharan Africa; the region currently with the highest proportion of people at risk, due to changes in climate. This review not only suggests the need to develop an efficient and effective model that will predict *Schistosoma* spp. population dynamics but seeks to evaluate the effectiveness of several current control strategies. The design of a framework model to predict and accommodate the future incidence of schistosomiasis in human population dynamics in sub-Saharan Africa is proposed. The impact of climate change on schistosomiasis transmission as well as the distribution of several freshwater snails responsible for the transmission of *Schistosoma* parasites in the region is also reviewed. Lastly, this article advocates for modelling several control mechanisms for schistosomiasis in sub-Saharan Africa so as to tackle the re-infection of the disease, even after treating infected people with praziquantel, the first-line treatment drug for schistosomiasis.

## 1. Introduction

Neglected tropical diseases (NTDs) are a group of chronic infectious diseases that have been ignored for several decades but which typically affect poor people who live on wages below US $2 per day, particularly those living in poor rural areas as well as destitute urban regions with limited access to public health facilities, which are prominent in sub-Saharan Africa (SSA) [1,2,3]. Over 500 million people in this region bear the full impact of these group of diseases which have a significant influence on the health, social, financial and economic status of both governments and households. According to the 10th meeting of the World Health Organization (WHO) Strategic and Technical Advisory Group for NTDs in 2017, the number of NTDs has been increased to 20 following the addition of three diseases namely, scabies and other ectoparasites, snakebite envenoming, and chromoblastomycosis and other deep mycoses. The existing 17 are Buruli ulcer, Chagas disease, dengue and Chikungunya, dracunculiasis (guinea worm disease), echinococcosis, foodborne trematodiases, human African trypanosomiasis (African sleeping sickness), leishmaniasis, leprosy (Hansen’s disease), lymphatic filariasis, onchocerciasis (river blindness), rabies, schistosomiasis, soil-transmitted helminths (STH) (*ascaris*, hookworm, and whipworm), trachoma, taeniasis/cysticercosis and yaws.

Schistosomiasis, also known as bilharzia, is identified as the second most widespread NTD in SSA after hookworm infections [4]. It is a parasitic infection which results in acute and chronic disease caused by trematode flatworms of the *Schistosoma* genus. The disease has dire consequences on child development, agricultural productivity, and outcome of pregnancy in affected regions [5]. In Africa, there are *S. mansoni* (intestinal), *S. haematobium* (urogenital), *S. intercalatum* (intestinal), and *S. guineensis* (intestinal) [6,7]. The first two are common and widespread, while the last two are rare and restricted to central African countries. *Schistosoma japonicum* is predominant in China and a few other areas, but its levels are low. *Schistosoma mekongi* is rare and associates with the Mekong River [8,9,10].

The life cycles of all the *Schistosoma* spp. are all similar yet very complex as the parasite alternates between two hosts: the intermediate (snail) and the definitive (such as human, bovines and domestic cattle) host [3,6]. In the human hosts, the infection is associated with several diseases such as malaise, skin rashes, fever and abdominal pain in an acute situation, while the chronic stage of the disease may result in liver, lung, intestinal or urogenital diseases, which are highly dependent on the species one is infected with [11]. Reoccurrence of infection over the years may additionally cause cancer of the bladder, pulmonary hypertension, and blockage in the urinary tract that may ultimately lead to other related complications and even death. In recent times, several studies [12,13,14,15,16,17] have deduced that climate and the changes associated with it such as temperature change, precipitation, humidity, flooding, salinity and drought have adverse effects on the *Schistosoma* life cycle, especially in sub-Saharan Africa. Hence, understanding the relationship between the variability in climate change and the life cycle of the schistosome worm will be a promising and predictive target in the control of schistosomiasis in the sub-Saharan Africa region.

## 2. Impact of Climate Change on NTDs and Human Health in Sub-Saharan Africa

The progression or increase in population dynamics of various health-related issues in SSA can be linked to a number of climatic variations, which include changes in temperature, rainfall or precipitation, air movement, drought, flooding, salinity and others [12,16,17,18,19]. According to the 2013 Policy Brief Report of the African Climate Policy Centre (ACPC) on the issues of climate change and health in Africa, it was stated that climate change in SSA will not only present health consequences but will also have adverse effects on the economic growth and development of certain SSA countries, while mounting additional impact on environmentally related issues [20]. This is due to the vulnerability of SSA countries to climate change as compared to other continents in the world. This susceptibility is due to the existence of high levels of poverty, armed conflict and weak institutions in SSA which limit the capacity of African countries to deal with the additional health challenges posed by climate change [20]. It was further averred that the level and category of health-related issues caused by climate change in SSA is significantly different from one community or region to another. This diversity is as a result of microclimate differences, geographic and socio-economic factors, accessibility and availability to quality health infrastructure, as well as underlying epidemiology and communication capabilities [3]. The influence of climate change in human health can either be direct or indirect; such changes include elevated levels of mortality and morbidity, disease patterns and incidence such as cancer of the skin, thermal stress, eye diseases, allergic disorders, cardio-respiratory diseases, malnutrition, inaccessibility to food and water, famine, droughts, infectious diseases due to migration, and the increase in population dynamics of vector and water-borne diseases [21,22,23]. The most important impacts of climatic change on human health include an increase in the prevalence of NTDs, malaria, malnutrition, diarrhea, and meningitis [23].

Several studies have confirmed the impacts of climate change on the incidence of cholera in SSA [24,25,26]. In an annual study carried out in Lusaka, Zambia by Fernández and co-workers on the influence of climate change on the dynamics of cholera, it was observed that the number of people infected with cholera increased by 4.9% due to increased levels of rainfall for three weeks, due to an increase in temperature levels for six weeks prior to the beginning of the rainy season. It was therefore posited that this region might be further confronted with an increase in the number of cholera cases within the following three weeks of rainfall [24]. In another related study carried out in Kwazulu-Natal, South Africa by Mendelsohn and Dawson, it was observed that an increase in the sea surface temperature and precipitation caused by climate change correlated with the prevalence of cholera in the area [27]. In a model study done in Tanzania, Trærup and co-workers [28] incorporated historical data on climate change and the incidence or burden of cholera together with socio-economic data as the control. It was observed that a 1 °C increase in temperature significantly increased the risk of cholera by 15–29%. Based on their model, the authors projected that an increase in temperatures between 1–2 °C will increase the prevalence of cholera with about 0.32–1.4% in Tanzania by the year 2030 [28].

In addition, the malaria burden in SSA has also been attributed to climatic change; it was observed that the increase in endemic transmission and spread of malaria in disease-free regions correlated with an increase in temperature between 32–33 °C [29]. On the other hand, the spatial distribution of anopheline larval habitats in the western Kenyan highlands correlated with change in topography and the effects of land cover types [30]. In another model study carried out by Ngarakana-Gwasira and co-workers [31], it was observed that an increase in both the reproduction and transmission dynamics of the falciparum parasite correlated with a change in climatic factors such as rainfall and temperature. It was averred that the malaria burden is likely to increase, not only in East Africa but also in the tropics and the highland regions of SSA, while a reduction in *P. falciparum* malaria will be experienced in Northern Africa [31]. A recent study carried out in the KwaZulu-Natal Province, South Africa observed that the increase in transmission of *Anopheles arabiensis* is dependent on seasons and that the increase in transmission of the malaria vector additionally depends on climatic factors, such as temperature and rainfall [32].

Other studies in which the effects of climate change have been examined on health-related issues include the prevalence of meningitis in North-west Nigeria [33], the increase in cerebrospinal meningitis in Ghana, as well as the threat of diarrheal disease in Botswana [34,35]. Therefore, climate change should also be seen as a potential factor that enhances the increase in the population dynamics of schistosomiasis in SSA. Transmission of this infection relies on the presence of an intermediate host in the form of freshwater snails. Thus, the following sub-topics will expand more on the effects of climate variability on snail fecundity, production, and mortality.

## 3. Snail as an Intermediate Host in the Transmission of Schistosomiasis

Two hosts are known to be important for the spread of schistosomiasis; the intermediate and the definitive host. The asexual phase of the reproduction cycle of the parasite takes place within the snail, while the sexual phase takes place in humans. Within the definitive host are generations of fertilized schistosome eggs which are released through human feces or urine into freshwater where they hatch into free-living ciliated organisms called miracidia; this is the first larval phase of the parasite, with the ability to penetrate and infect aquatic snails that serve as the intermediate host to the parasite. Once in the intermediate host, the miracidia continue their lifecycle to produce multiple cercariae, which is the second larval stage of the parasite. Cercariae have the ability to penetrate and infect human skin once in contact with infested waters.

In the life cycle of the *Schistosoma* worm, four genera of snails have been identified to serve as intermediate hosts for the parasite which include *Bulinus*, *Biomphalaria*, *Tricula*, and *Oncomelania*. These snails’ species can be sub-divided into two groups based on their habitats: *Bulinus* and *Biomphalaria* snails, also known as aquatic snails, have the ability to live and survive underwater but generally cannot endure and survive elsewhere for longer periods of time, while *Oncomelania* and *Tricula* snails can adjust and survive both within and outside water [36,37]. *Oncomelania* and *Tricula* are important for the transmission of *S. japonicum*, *S*. *sinensium* (only in rodents), *S. megonki*, *S. malayensis*, and some other *Schistosoma* infections in humans and other animals in Asia, most especially in China, Philippines, Malaysia, and Thailand, but they are not present in Africa. *Oncomelania* also serves as intermediate hosts in the transmission of *S. japonicum* in Japan and the Sulawesi region of Indonesia [38,39].

### 3.1. Distribution of Bulinus Species in Sub-Saharan Africa

*Bulinus* is a group of freshwater snails of the gastropod genus, which belongs to the Planorbidae family. It is composed of four groups of species: *B. africanus*, *B. forskalii*, *B. reticulatus*, and *B. tropicus* (or *B. truncatus* complex), which are all sub-divided into 37 species with some species responsible for the transmission of *S. intercalatum* and *S. haematobium* larval parasites. There are nine species of *Schistosoma* transmitted by *Bulinus*, three that infect humans and six that infect Bovids or rodents [40], *Bulinus* can survive outside freshwater as they can aestivate [41]. In SSA, over 112 million cases of urogenital schistosomiasis are caused by *S. haematobium*, which accounts for about 50% of the total incidence of *Schistosoma* infection in this region [5,42]. This may be largely due to the wide geographical distribution of the *Bulinus* spp. (its intermediate host) in this region with the snail mostly endemic to Cameroon, Egypt, and Senegal among others [4].

The transmission of urogenital schistosomiasis within and between SSA countries is significantly different from one region to another and is dependent on the functions performed by the various *Bulinus* species within one ecological region to the other [43,44]. For instance, in Senegal, *B. truncatus* and *B. globosus* are the main hosts, while it has also been thought that *B. senegalensis* and *B. umbilicatus* are able to transmit the parasite [43,44]. In Cameroon, the transmission of this disease predominantly involves *B. truncatus*, while in Southern Africa, *B. africanus* and *B. globosus* have been recognized to be responsible for the transmission of *S. haematobium* and a less-recognized species *S. mattheei* [45,46]. The disparity in the role of the *Bulinus* species in the transmission of urogenital schistosomiasis was highlighted in a recent study by Zein-Eddine and co-workers [47]. In this study, significant differences were observed in the genetic makeup of seven *Bulinus* spp. examined within three areas of high prevalence of *Bulinus* spp. in sub-Saharan Africa (Cameroon, Egypt, and Senegal). Added to this, the observable genetic diversity in *Bulinus* spp. may be indicative of the roles exhibited by the snail species in the transmission of schistosomiasis in different regions of SSA [47].

### 3.2. Distribution of Biomphalaria Species in Sub-Saharan Africa

*Biomphalaria* belongs to the genus of freshwater gastropod snails, which are members of the Planorbidae family. They are otherwise known as Taphius and serve as intermediate hosts for the transmission of *S. mansoni* infection leading to intestinal schistosomiasis. *Biomphalaria* snails are usually found in tropical freshwater ponds within sub-Saharan Africa and South America or in subtropical regions within a 30° latitude radius [48]. In a natural setting, *Biomphalaria* species cannot survive outside the freshwater [49].

There are several existing *Biomphalaria* species which are known vectors in the transmission of intestinal schistosomiasis worldwide; 22 of which have been recognized in America, while 12 dominant species have been discovered in SSA [50,51]. Egypt, particularly, houses two special hybrid species: *B. alexandrina* and *B. glabrata* [52]. Currently, four other species—*B. pfeifferi*, *B. tenagophila*, *B. straminea* and *B. glabrata*—recently extended their natural homes to some SSA countries and subtropical regions such as Egypt, Congo, Florida, Texas, Hong Kong, and Louisiana [48]. The movement and invasiveness of biomphalaria to another region may be due to climate change and human activities, such as the trade of aquarium plants. More so, human activities can also facilitate the spread of snails through freshwater environments such as canals, irrigation schemes, and hydroelectric dams. It is also likely that the creation of artificial freshwater habitats in urban and peri-urban areas, and the degradation of water quality may facilitate the expansion of the snail species [48].

Studies have shown that the *Biomphalaria* species can reside in slow-moving and minimal wave-acting waters, and in some cases, prefer to stay in ponds or pools where they exhibit a high degree of tolerance to changes in temperature. This condition seems suitable for the production of miracidia, which seek out and penetrate these snails to undergo asexual reproduction to form cercariae [53,54]. There are many Biomphalaria species in places like Lake Victoria in Uganda where high transmission takes place [14]. In sub-Saharan Africa, studies have shown that the wide geographical distribution of *S. mansoni* is related to the presence of *Biomphalaria* snail species in the region [55,56].

## 4. Impact of Climate Change on Schistosomiasis Transmission in Sub-Saharan Africa

Climate variability has been predicted as a potential factor that may influence the transmission of schistosomiasis [18,19,57,58]. Variations in the weather conditions have been recognized to have a significant impact on the lifespan (mortality) and fecundity rate of both snails and worms transmission during the schistosome life-cycle [59,60]. Hence, the role of climate change factors such as alterations in temperature, rainfall/precipitation, flood, drought, and pH among others as depicted in the model shown in Figure 1 and will be discussed in the following sections of this article.

### 4.1. Effect of Changes in Temperature on the Intermediate Host in Schistosomiasis Transmission

It is believed an increase in global temperature has a pronounced effect on the interactions that take place among organisms in an ecosystem [22,61]. An increase in temperature has been suggested to influence the complex interactions that occur between schistosomes and their snail intermediate host. As shown in Figure 1, the exact effect of an increase in temperature in SSA on the *Schistosoma* snail intermediate host of the parasite may influence growth, distribution, survival and fecundity rate, as well as make unfavorable breeding conditions for both freshwater snails and the schistosomes themselves, which may, in turn, affect the population dynamics of *Schistosoma* infections in SSA depending on snail types and schistosome species present in the area. It has been shown in a number of studies on the eradication of schistosomiasis in certain parts of SSA, that an increase above optimal temperature (26–31 °C) can eventually lead to a decrease in mortality rates of snails [12,13,19,62,63]. Additionally, an increase in temperature level may decrease the infectious stage of Schistosoma parasite due to a decrease abundance in snail production and a decrease in the growth and developmental rate of the parasite.

The rate at which the miracidia penetrate snails, as well as the release of cercariae into the larval stage of the parasite and its penetration into the skin of the definitive host, are temperature-dependent [63]. The production of cercariae within the intermediate host is assisted by a higher temperature of about 15 °C to 31 °C, and this does not only help the production of cercariae but also plays a significant role in increasing the metabolic activity, energy, and vitality of the snail to intensify the rate of cercarial production within the snail [19,60]. Studies have shown that the fecundity, survival, and mortality rate of several intermediate hosts of schistosomes and the developmental rate of the worm within the host, acclimatization or adjustment to a particular environment were dependent on a change in temperature [12,49,62,64,65,66,67].

McCreesh and Booth [63] developed a life cycle-based model of temperature-sensitive stages of *S. mansoni* and the life cycle of the *Bi. pfeifferi* snail. It was discovered that the number of snails approximately remained constant between 15–31 °C; any temperature outside this range saw a steep decrease in the number of intermediate hosts. This implies the snail population lacks the ability to survive beyond this temperature range. More so, the highest at-risk temperature range for infection with schistosomes in calm waters for humans was shown to be 16–18 °C, around 1 pm and between 6 pm–10 pm. However, the highest risk of infection in flowing waters was determined to be between 20–25 °C, between 12 pm–4 pm according to the study, thus, this assertion needs further elucidation to be well understood. On the whole, the authors showed the risk of infection increases suddenly as the temperature increases beyond the minimum degree required for continued transmission.

More so, Appleton and Eriksson [68] investigated the role of temperature fluctuations above optimal levels and the effect on the fecundity of *Bi. pfeifferi,* the snail species responsible for *S. mansoni* transmission. They observed the fecundity rate for freshwater snails decreased drastically as the temperature increases above 27 °C, which is believed to be the optimal temperature for snail fecundity [68]. In the same vein, at temperatures above optimal level, snails with shell height ranging between 1.5–2.5 mm failed to produce multiple eggs. Moreover, the survival rate of the snails reduced drastically, which in turn affected the production of the *S. mansoni* cercariae.

### 4.2. Effect of Changes in Rainfall on the Intermediate Host in Schistosomiasis Transmission

The effect of changes in rainfall/precipitation on population dynamics or in the transmission of schistosomiasis in SSA cannot be over-emphasized. It is believed that an increase or a decrease in water levels occur because of rainfall patterns which influence the transmission of schistosomiasis. A quantitative analysis of change in the inter-annual total rainfall of Ga District in Ghana showed the total rainfall and number of rainy days positively correlated with the prevalence of schistosomiasis in the area [69]. Furthermore, it was shown that years with reduced amounts of rainfall correlated with a low prevalence of schistosomiasis, while years with moderate as well as high rainfall correlated with a high prevalence of schistosomiasis. However, no significant association was made between the amount of rainfall and the disease itself [69].

In another related study conducted in Ethiopia by Xue et al. [17], it was established that rainfall is a prominent climatic factor responsible for the increase in population dynamics of schistosomiasis through the accumulation of sufficient surface water in ponds. An increase in rainfall promotes and provides a breeding space for the snail population due to an increase in the volume of runoff waters that are channeled through irrigation canals. This may, in turn, increase flow velocities, thereby promoting contact between the parasite and its intermediate host. An increase in water levels due to high rainfall may also cause water turbulence which may increase the flow rates of water that, in turn, disturb snail habitats as well as the decreased survivability of cercariae [17].

In years past, the considerable differences in the distribution of the *Bulinus* species in certain areas were observed to depend on the amount of rainfall, the periods of dry and rainy seasons, as well as the intervals between these seasons [70]. During prolonged dry periods, there is an observable ‘dry-out’ in the natural habitats of snails, which may, in turn, result in the death of those that host the parasite [57]. In SSA, the distribution of the *Schistosoma* species varies depending on the amount of rainfall. This was observed in a study conducted in three villages in the lower delta region of Senegal by Ernould and co-workers [71]. It was discovered that during rainy periods, there was a drastic increase in the transmission of *Biomphalaria pfeifferi*, the prospective intermediate host of *S. mansoni* when compared to *Bulinus globosus*, the intermediate host of *S. haematobium*; this suggested an increase in the transmission of *S. haematobium* during the dry period [71].

### 4.3. Effect of Flooding on the Intermediate Host in Schistosomiasis Transmission

Some region in SSA may experience an increase in the transmission of schistosomiasis due to an increase in flooding caused by temperature variability or changes in weather patterns, which has been suggested to have a serious impact on human health [72,73]. During flooding, a huge number of people encounter contaminated water resulting in infection with the schistosome parasite [16,18]. This assertion was supported by the observation of Wu and co-workers [74] who elucidated that the dispersal patterns of intermediate snail hosts with respect to acute and chronic infections caused by the parasite in the Peoples Republic of China was related to flooding. It was observed that the habitats of snails present during the years when flooding occurred were 2.6–2.7 times larger than in those years when water levels were normal. Additionally, the re-emergence of snails was observed due to floods in habitats where the snail population have been previously reported to be eliminated. According to Wu and his colleagues, both the density and infection rate of snails infected with the parasite dropped during the first two years after the floods, while in the third year, a significant increase was observed in the infection and density of snails to the parasite [74]. This observation was supported by results from mathematical models created by Longxing and co-workers [16] in their study conducted in Anhui Province of the Peoples Republic of China that showed how floods act on the stability of an endemic equilibrium. The numerical simulation and generated data revealed there would be an extremely serious schistosomiasis outbreak in the study area about three years after the flood. It was further averred that there was a marked increase in the number of acute schistosomiasis cases in the years characterized by the floods, with an average increase of about 2.8 times more cases as compared to years with normal water levels [74].

It would, therefore, be safe to conclude that due to the increase in flooding activities in the SSA countries, there may eventually be an increase in the occurrence of new habitats for both snails and schistosomes, as well as reoccurrence of snails in some areas where they have previously been reported to be eliminated.

### 4.4. Effect of Drought on the Intermediate Host in Schistosomiasis Transmission

Drought is a climatic factor with a significant impact on the socio-economic and health-related issues of an individual and the society at large. Some of these effects include lack of water and food, increase in diseases caused by airborne particles such as smoke and pollen, valley fever caused by fungi, germy hands, mosquito-borne diseases, as well as recreational injuries among others [75,76]. This type of climatic factor may have future effects on the transmission of schistosomiasis in SSA because certain snail intermediate hosts such as *Oncomelania* which is not found in Africa, actually have the ability to withstand and survive dry environments for extended periods of time due to their operculum, which is capable of closing shell during periods of drought for as long as 2–4 months [13,77,78]. However, prolonged drought in some countries like Nigeria, Ethiopia and Zimbabwe, has resulted in a remarkable reduction in the prevalence of schistosomiasis due to a decrease in the reproductive and survival rate of snails responsible for transmission of the parasite in those areas, as well as the decrease in the transmission sites [56,79,80,81].

Studies have shown that during aestivation, uninfected *Bulinus* snail species have the ability to withstand longer periods of drought than infected snails [56,57,82]. This implies that during drought, infected snails die off and this alters the transmission of schistosomiasis in drought-stricken areas [15,56,57,82]. Mutuku and co-workers [56] analyzed temporal changes on the spatial transmission pattern of *S. haematobium* worms on different age groups and their relationship to ponds infested with *Bulinus* snails in coastal Kenya. It was observed that the ponds dried up and hence, there were no sources of infection; very few or none of the snails that infested the ponds in the past were detected during the major drought between 2001 and 2009. They further averred that the hydrological changes and the long-term drought resulted in the absence of *Bulinus* snails nine months after the pond had been refilled [56].

In another related study by Senghor and co-workers [15], the impact of drought on the snail intermediate hosts of urogenital schistosomiasis in Niakhar, West-Central Senegal was studied from July to November–December in 2012 and 2013 respectively. It was observed that out of the two *Bulinus* species (*B. senegalensis* and *B. umbilicatus*) collected in the study area, only *B. senegalensis* was found in all 17 sampled sites, while *B. umbilicatus* was found in only one out of the 17 surveyed sites. In the years 2012 and 2013, a total number of 1032 and 8261 *B. senegalensis* were collected, respectively, while a total of 901 out of the 1032 and 6432 of the 8261 snails were tested for *Schistosoma* spp. infection within the stipulated time periods. On the other hand, a total number of 58 and 290 *B. umbilicatus* snails were also collected during the years 2012 and 2013, respectively, while a total number of 58 and 281 snail species were tested for infection within the respective designated time periods [15]. Overall cercarial shedding of 0.0% and 0.12% was observed for *B. senegalensis*, as well as 13.79% and 4.98% for *B. umbilicatus* during these years. It was further posited that individual *Bulinus* species, ranging between 7–9.9 mm in size, were present in the month of July; 63.6% and 57.8% for *B. senegalensis* and *B. umbilicatus*, respectively. For the first time, this study reported that *B. umbilicatus* can maintain *Schistosoma* larvae for as long as seven months of drought, thereby resulting in the transmission of schistosomiasis in early July leading to an increased risk of schistosomiasis transmission in the study area [15]. Therefore, it is safe to deduce that prolonged drought periods for more than nine months can stop the transmission of schistosomiasis at the foci, while drought periods lasting less than seven months can aid the transmission of the disease in other parts of the region due to maintenance of the larval stage.

### 4.5. Effect of pH and Conductivity on the Intermediate Host in Schistosomiasis Transmission

Physico-chemical parameters such as pH and conductivity in relation to climatic factors are known to have a considerable influence on the population dynamics of schistosomiasis transmission in SSA. In simple terms, pH is the presence of hydrogen ions in water or soil and the amount of acidity in water is measured by the pH. The acidity or pH of water differs greatly from one region to another and is highly dependent on climatic change. Due to global warming, there is a strong release of carbon dioxide into the atmosphere by the ocean leading to a decrease in pH of about 0.28–0.7 U, which influences the acidity of the ocean with a pH increment between 7.4–7.8 [83]. An increase in precipitation is expected to result in the acidification of the atmosphere or the oceans, which can pose a serious threat to marine organisms through tropic interactions and biodiversity caused by calcification [84,85,86].

This type of environmental condition has been shown to influence the population dynamics of several diseases that include the infection and lysogenizing of recipient cells or hosts by *Vibrio cholerae* [25]. There are only a few studies that have shown the influence and interaction between change in pH and its resultant effect on parasites and their hosts. In an extensive study conducted by Koprivnikar and co-workers [87] on the effects of temperature, salinity and pH on the rate of survival and activity of marine cercariae, it was discovered that neither species of the cercariae was affected by pH alone but there were associations with salinity and time. This was observed in a multivariate test which showed a significant decrease in the number of cercariae through the time period. The study further showed the survival and active rate of cercariae only occurs at a high pH of 8.2 [87].

Other studies that have been conducted elucidating the nexus between change in pH and the intermediate host of schistosomiasis, including one carried out in Uganda by Rowel and co-workers [14]. It was shown that physico-chemical factors such as pH and conductivity do influence *Biomphalaria* populations and infections. A disparity in the population dynamics of *Biomphalaria* from one location to another was observed but depended on the lake and the pH. In both Lake Albert and Lake Victoria, a positive relationship between the *Biomphalaria* population and infections was observed, while in Lake Albert, a negative relationship with higher salt content was observed. It was further stated that out of all the *Biomphalaria* snails collected from Lake Albert, 8.9% were infected with digenetic trematodes, 15.8% shed *S. mansoni* cercariae, while 84.2% were infected with non-human-infective trematode. On the other hand, from Lake Victoria, 2.1% of all collected *Biomphalaria* snails were infected with digenetic trematodes with 13.9% of them shedding *S. mansoni* cercariae, 85.7% shedding non-human-infective cercariae, and 0.4% of the infected snails shedding both types of cercariae [14].

In a related study by Marie and co-workers [88], the physical and chemical properties of water quality on the density and distributions of some freshwater snails collected from eight different streams within the Egyptian governorate were investigated. Within these eight streams, there was an observable production of snail species at a wide-water pH range with the highest percentage ranging between pH 7.6–8.5, which is similar to that obtained by Ntonifor and Ajayi [89], which showed that the pH ranges that harbour snail production falls between pH 7.2 and 10.9. A fluctuation in the pH below or above 7.0 and 9.0 results in a decrease in snail production from 68.94% to approximately 1.35% and 13.28%, respectively [88]. It was further observed that *B. alexandrina* snails were the most enumerated snail species with the ability to tolerate and adapt to various environmental circumstances. This concurs with the work of Kazibwe and co-workers [90] where they observed that the abundance of *B. sudanica* negatively correlated with pH levels. Other previous studies have shown the abundance of *B. alexandrina* and *B. truncatus* snails at a pH of 6.0–6.5 and pH of 6.9–7.2 [91,92]. Unfortunately, no study has examined the effects of pH and conductivity on the production, survival and fecundity of *Bulinus, Oncomelania* and *Tricula*.

### 4.6. Effect of Salinity on the Intermediate Host in Schistosomiasis Transmission

Salinity is another important factor influence by climate change which have considerable effects on schistosomiasis transmission. The variability in climatic factor caused by global warming can result in the elevation of sea levels which can, in turn, raise the salinity of water bodies in coastal regions [93]. It was hypothesized by Ramasamy and Surendran [93] that an increase in the level of saline water bodies in coastal regions correlates to a rise in sea levels leading to an elevation in the densities of salinity-tolerant vectors, which in turn result in the adjustment of freshwater vectors (snails) to breed in saline/brackish waters. Among the studies that have focused on the impact of salinity on parasites, it has been shown that cercariae possess a high tolerance to salinity fluctuation and their production from the intermediate host can be influenced by it [41,94]. A decline in cercariae production occurs as salinity concentration reduces. Moreover, the overall impact of high salinity has been associated and is more favorable towards schistosome parasites rather than their intermediate snail hosts [95,96,97]. Thompson [98] and Paraense [99] observed *B. straminea* and *B. glabrata* freshwater snails have the ability to survive in saline water of about 5.0ppt in coastal regions. It was further observed that *S. mansoni* eggs from these coastal regions have the ability to hatch in saline water of 6.0 ppt [100]. Mostafa [101] carried out a study to ascertain the effect of salinity on the survival of *Biomphalaria arabica*, the intermediate host of *S. mansoni* in Saudi Arabia, from freshwater bodies by collecting and exposing the snails to NaCl concentrations of 1%, 2%, 3%, 4%, 5%, 6%, 7%, 8%, 9%, 10%, as well as a series of concentrations lying between the one that produced 100% mortality and the preceding one. Results from this study showed *B. arabica* snails remain alive in 5% NaCl concentration and at 7.2% NaCl concentration, 100% mortality occurred. He further averred that in the presence of water lettuce, *B. arabica* showed pronounced resistance to an increase in salinity, which may account for the abundance of *B. arabica* in the study area. This finding was supported by the study of Neto and co-workers [102] in which the effects of biological, physical, and chemical factors on breeding sites in Porto de Galinhas, Brazil was studied. It was shown that salinity indices up to 1003 d, is non-infective on the snail species and the *S. mansoni* strain, which are highly adapted to coastal environments with high salinity.

## 5. Effects of Climate Change on the Geographic Habitat and Behaviour of Snail Species

In sub-Saharan Africa, changes in weather patterns can have a negative impact on environmental health risks, as well as pose serious negative effects on disease management among the people. For instance, infectious diseases like schistosomiasis and malaria have largely been controlled over the years by mass drug administration and environmental management in the absence of vaccines. However, complications such as changes in ambient temperature, temperature ranges, changes in precipitation, and water flow can influence the geographical habitat and behavioral conditions of vectors (snail species) thus, making it more difficult to deal with the disease.

In South Africa, *Bulinus* africanus is majorly found in the eastern half of the country as far south as the Kromme river in Humansdorp, while *B. globosus* is limited to the extreme eastern parts of Mpumalanga and Limpopo provinces and to a small area of north-eastern KwaZulu-Natal. Both the *B. africanus* and *B. globosus* are vulnerable to dryness and their intrinsic rate of increase is considered relatively low [46] but is strongly linked to temperature. It has been reported that the distribution of the *Bulinus* group across South Africa appears to be determined greatly by temperature and water flow rate or body [103]. *Bi. pfeifferi* which is generally found in slow or still moving permanent water bodies, is particularly vulnerable to desiccation and thus, is unlikely to be found in temporary rain-filled habitats. Its distribution across South Africa is similar to that of *B. africanus* but does not extend further south than Port St Johns. Similar to the *Bulinus* species, this snail’s distribution is mostly influenced by water body type and temperature [104].

De Kock and Wolmarans [105] investigated the geographical distribution and habitats of the *Bulinus* species in South Africa. The results from their decision tree analysis showed that temperature is the most important determinant factor in the geographical distribution of the *B. africanus* group in South Africa, with over 70% of the samples examined for this group recovered from habitats with either slow-flowing or standing water [46]. This supported an experimental and field study that documented that *B. globosus* is sensitive to temperature gradients and capable of seeking out parts of the habitat where temperatures are nearest to its optimum [104].

It has also been shown in another experimental study carried out by De Kock [105] that there was a marked difference between the optimum temperature ranges for survival and the optimum range for reproduction for a particular species, and that these ranges could differ even for the same species depending on their specific area of origin. This supported the view by Brown [106] which showed that *B. africanus* is more capable of colonising habitats under cooler climatic conditions than *B. globosus* [107] because it has also been reported that *B. africanus* survived longest at constant low temperatures [39], while *B. globosus* survived longest at constantly high temperatures [39]. It has also been reported by Appleton and Stiles [107] that the entire endemic area of the snail species for schistosomiasis in South Africa is over rock formations which are resistant to erosion. While accepting that waters flowing over such rocks tend to provide permanent pools that serve as refuges for snails, Brown [108] is of opinion that this effect is no more than locally significant and points to the fact that large parts of the area from which these snails are absent contain apparently suitable habitats.

These views were further corroborated by a De Kock and Wolmarans study where a lower temperature index (3.136) was reported for *B. africanus* when compared to that of *B. globosus* [46]. This presents a logical explanation for the ability of the *B. africanus* species to colonise habitats on the highveld of the Free State, North-West and Gauteng Province of South Africa. The precarious existence of populations of *B. africanus* under cooler conditions on the fringe of its geographical distribution is attributed to the relatively low values realized by cohorts of this species, even at optimal temperature regimes in life-table experiments [105].

Salinity is another important environmental factor that can influence the geographical distribution and habitats of snail species in South Africa. De Kock and Wolmarans [46] calculated the effect of salinity of the water on the habitat of the *Bulinus* species and it was indicated that this factor may play a relatively unimportant role in the distribution of this group in South Africa [109,110,111,112,113]. This supported the statement of Pretorius and co-workers [109] that a natural population of *B. africanus* declined when the salinity level is increased.

In [114], Appleton and co-workers performed a retrospective analysis of a well-documented prevalence of human intestinal schistosomiasis which occurred at Maun in the seasonal part of the Okavango Delta, Botswana. The prevalence of intestinal schistosomiasis in this region corresponded with the annual flow records of floodwater from the delta along the Thamalakane River at Maun, building the prevalence of the disease from very few cases in the 1950s and early 1960s to a peak prevalence of >80% in the 1980s. They further observed that the decrease in transmission was coincident with a period of lower discharge into the Thamalakane River and the establishment of a chemotherapy-based control programme. They posit further that the ephemeral nature of the Thamalakane River (dryness for long periods) is not an optimal habitat for *Bi. pfeifferi* and that it has to be re-colonized by this species after each drought period. This process of re-colonization relies on snails being carried into the river from the permanent delta during a flood. It will then take several years for new foci of *S. mansoni* transmission to become established and reach the stage where severe morbidity will appear, and intestinal schistosomiasis can again be recognized as a public-health problem in that region [114].

## 6. Role of Mathematical Modelling in Disease Epidemiological Studies

Mathematical modelling is a system used in describing real-life problems in both mathematics language and concepts. Mathematical modelling comprises several methods which include statistical models, dynamical systems, theoretical models or differential equations [115]. The applications of modelling have been proved to be very useful in natural sciences such as biology, meteorology, and earth sciences among others. Additionally, its role in the field of engineering, social sciences, medicine, physical system control and risk management cannot be over-emphasized [116,117,118,119]. Additionally, mathematical modelling plays an important role in the prevalence of infectious diseases, where it is useful in investigating or examining and quantifying the effect and cause of the spread of infectious diseases [120,121]. Modelling can also be helpful in decision-making due to the projected results generated such as changes in the pattern of disease spreads due to interventions [122].

Mathematical modelling has been used in the People Republic of China (PRC) to determine the effect of multi-component integrated approaches in eliminating schistosomiasis [123,124,125]. Meanwhile, Africa Li and colleagues [126] carried out a model-based study which adapted an established *schistosoma* infection transmission model that couples local human and snail populations, and includes part of snail ecology and parasite biology. In the model, they employed data from lower-risk, moderate-risk, and high-risk rural villages in Kenya, and the simulated control was carried out via MDA. The model also compared 2012 WHO guidelines with a modified adaptive strategy that tested a lower-prevalence threshold for MDA and shorter intervals between implementation, evaluation, and modification. They further explored the addition of snail control to the modified strategy. The findings of the modified adaptive strategy for MDA, without or with snail control, will be more effective than the current WHO guidelines from 2012 in achieving key public health goals, especially in high-risk communities that emerge as persistent hotspots of *schistosoma* infection.

Yang and Bergquist [127] employed the combination of two meteorological models (Coupled Model Intercomparison Projects (CMIPs) and Representative Concentration Pathways (RCPs)) alongside with biological modelling to investigate the replication and survival of the snail intermediate host as well as the maturation of the parasite stage within the snail at different ambient temperatures. In their study, the potential geographical distribution of the three major schistosome species (*S. mansoni*, *S. haematobium* and *S. japonicum*) was examined with reference to different transmission capabilities at the monthly mean temperature, the minimum temperature of the coldest month(s), and the maximum temperature of the warmest month(s). Based on their study, it was shown that there will be an increase in transmission areas for all three species in 2021–2050 and 2071–2100. However, the model also gave room for potential reductions in the transmission of schistosomiasis in certain areas [127].

Pedersen et al. [128] showed that snail habitat suitability is highly variable in Zimbabwe, with distinct low- and high-suitability areas and that temperature may be the main driving factor. This was observed in a biologically based model for the freshwater snails *B. globosus*, *Bi. Pfeifferi*, and *Lymnaea natalensis* when major potential climatic and environmental drivers predicted to be suitable for snail habitat were incorporated into the model. The findings of their study showed that future climate change in Zimbabwe may cause a reduction in the spatial distribution of suitable habitat of host snails probably with the exception of *Bi. pfeifferi*; the intermediate host for intestinal schistosomiasis whose spatial distribution may increase as time goes by and decrease towards 2100 [128].

In [13], McCreesh and co-workers carried out a climate change model analysis which may be temperature-sensitive to the stages of *S. mansoni* and intermediate snail host in Eastern Africa. The model was run using low, moderate, and high warming climate projections of the region. The sensitivity of predictions to different relationships between air and water temperature, as well as different snail mortality rates were determined. Their findings predicted changes in the transmission of *S. mansoni* as a result of an increase in temperature over the next 20 and 50 years. The results further predicted that *S. mansoni* may spread to new areas outside the existing area where control programmes are underway [13].

In contrast, Kalinda and co-workers [129] used data from laboratory and field experiments to develop a deterministic compartment simulation model for the life cycle of *B. globosus*. Their model generated fecundity of snail and survival rates similar to those in the laboratory. The model also produced reasonable snail population dynamics under a change in weather patterns. It was also shown that there is decrease in number of both the snails and cercariae due to an increase in the environmental temperatures, with maximum reduction in abundance of snails by 14% and 27% and decrease in maximum cercariae by 8% and 17% when the temperatures were increased by 1 °C and 2 °C, respectively. Their findings suggested that a rise in future temperature due to climate change may alter the abundance of *B. globosus* and influence the prevalence of schistosomiasis [129].

Mangal and colleagues [19] modelled the effect of temperature on the worm burden and prevalence of schistosomiasis for optimal disease control strategy. It was observed that the burden of *Schistosoma* reached the climax at a temperature of 30 °C and drastically reduces when the temperature is raised to 35 °C. Therefore, it was concluded that the best stable temperature for the spread of schistosomiasis ranges between 20 °C to 35 °C and that the optimum temperature for the survival of *Schistosoma* parasite is 20 °C, which is the temperature that the parasite can survive at; this can be helpful in disease control. In another related study by McCreesh and Booth [63], the temperature-sensitive stage of *S. mansoni* and the life cycle of its *Biomphilaria pfeifferi* intermediate host were simulated. It was observed that the infection of *S. mansoni* in rivers and lakes was very high ranging between 15–19 °C and 20–25 °C, respectively. Meanwhile, the survivability of the snail reduces drastically outside the temperature range of 14–26 °C. In a like manner, an epidemiological model was developed by Ngarakana-Gwasira and colleagues [31] to improve the prediction of the influence of climatic factors on the population dynamics and disparity of schistosomiasis strength in Zimbabwe. The study observed that the best temperature for the transmission of schistosomiasis in that region ranges between 18 °C to 28 °C and the optimal temperature for schistosoma transmission was about 23 °C. Additionally, it was observed that the *schistosoma* infection in snails increases at 22 °C when compared to other temperatures like 20 °C and 25 °C and that the *schistosoma* parasite died when the temperature was raised to 30 °C [31]. Recently, mathematical modelling of temperature and rainfall influence on *Schistosoma* species population dynamics in South Africa was carried out by adopting a schistosomiasis sub-model, which incorporated climatic parameters (temperature and rainfall). The study was employed to examine the impact of climate variability on the transmission dynamics of schistosomiasis. It was observed in the study that climate change poses a great influence on the reproduction number of both schistosomes and freshwater snails [1].

This review suggests that all hands must be on deck so as to develop models capable of predicting the effects of change in climatic weather patterns on the transmission of schistosomiasis using several climate-based models coupled with several integrated mathematical modelling approaches, and includes snail ecology and parasite biology to analyze how temperature and rainfall affect schistosomes population size. The influence of ambient temperature on the development and the mortality rate of the *schistosoma* parasite also needs to be considered over SSA using mathematical modelling. In particular, the model needs to be used in examining the impact of climatic factors on the life cycle of schistosomiasis and the dynamics of both the snail and schistosomes population over the region. Additionally, the model can be used in developing time-dependent control strategies and choose the best control strategy in eradicating the transmission of schistosomiasis in SSA.

## 7. Control Strategies for Schistosomiasis

Several preventive and control measures that have been employed over the years in eradicating and lessening the disease burden include the control of freshwater snails by environmental factors such as temperature, the current speed of water, water chemistry, light and shade, as well as vegetation reduction [130,131,132,133]. Other preventive control measures via intermediate snail hosts include the use of molluscicides, which can either be chemical (bayluscide, derivative of N-p-substituted phenyl uracil-5-sulphonamide; anilofos, fenitrothion, eugenol, and thymol) or plants (*Solanum nigrum*, *Ambrosia maritime*, *Thymelaea hirsute*, *Peganum harmala* and *Callistemon lauceolatus*) as toxic agents against freshwater snails and the schistosomes [134,135]. In addition, biological control that involves the use of different organisms such as Trematocranusplacodon fishes (e.g., tilapia) and Cairinamaschata (e.g., muscovy ducks) as predators of freshwater snails [136]. The host-parasite relationship is another control measure used in controlling the disease burden by making the freshwater snails develop resistance to some schistosomes infection via the activation of cytokines such as TNF-α and IL-1 [137]. Another internal defense protein response in molluscan control involves the upregulation of granularin; a protein with phagocytotic activity against foreign particles [138]. Additionally, genetic control involving the changing of the strain of highly susceptible snails to non-susceptible ones and the subsequent release of these snails that are resistant to schistosomes into natural habitats such as freshwater has been shown to be successful [139]. In the human host, praziquantel remains the only effective control in treating schistosomiasis, however, it has been shown over the years by both in vivo and in vitro studies that schistosomes may develop resistance against the drug due to mass drug administrations (MDAs) and it is also well-documented its inability to kill immature worms in the human host. Unfortunately, all these preventive and control measures are time-consuming and require huge financial resources to be implemented. More so, these control measures are unable to prevent total eradication and reinfection with the disease, hence the need for an alternative control strategy.

Over the years, several studies have employed mathematical modelling in identifying the best control and cost-effective strategies in the treatment and management of infectious diseases. The impact of chemotherapy on optimal control of malaria dynamics with infective treatments, the introduction of infected immigrants, and the use of insecticides against mosquitoes was described in a mathematical model by Makinde and Okosun [140].

Zhang and co-workers [141] developed a partial differential equation of a schistosomiasis age-based system model in human hosts and its application in treatment strategies incorporating the human definitive host and the intermediate snail host. The study suggested that the strategies of controlling the disease focusing on the rate of infection in any age group may be the most successful strategy in schistosomiasis control. However, treatment costs or the development of possible resistance of the parasite to chemotherapy due to mass drug administration was not considered [141]. Liang and co-workers [142] using a calibration approach and a mathematical model integrating various field data, designed control strategies whose feasibility of characterizing site-specific schistosomiasis transmission in endemic villages of south-western Sichuan, China, showed that a viable control strategy is important in lowering schistosomiasis transmission and that the population dynamics can be reduced by focusing on snail control, chemotherapy, and egg control [142]. It is therefore important to develop time-dependent control strategies in order to ascertain the best optimal strategy for schistosomiasis control, such as vaccination campaign once one of the human schistosome vaccine (BILHVAX, or the 28-kDa GST from *S. haematobium*) in trials is successful and approved; sensitization of the public about the disease; environmental control through the infrastructural interventions; improvement in the existing anti-schistosomal drugs and the production of new drugs that can mitigate as well as abolish re-infection of the disease and; control of the snails species that are involved in the transmission of the disease.

## 8. Conclusions and Future Perspectives

Modelling is a valuable tool that is useful in studying the prevalence of infectious diseases and can be used to investigate and quantify the cause and effects of the spread of infectious diseases [120,121]. Huppert and Katriel [122] considered modelling as a useful tool for developing disease prevention and control measures because it has value in predicting the likely outcome of a population level in order to implement different control measures. Moreover, modelling can also be helpful in decision making with regards to epidemiological issues such as changes in the spread of disease patterns [122]. Added to this, modelling has also been used to tackle the influence of temperature and rainfall on the transmission dynamics of diseases such as the evolution of cholera epidemics in Lusaka, Zambia [24], dynamics of poliomyelitis outbreaks [143], as well as the outbreaks of smallpox in realistic urban social networks [144], different smallpox epidemics in England [145], and responses to a smallpox epidemic in France; considering uncertainty factors [146]. The technique has also been used to gain great insights into the transmission dynamics of tuberculosis [147], as well as the transmission and bio-control of trypanosomiasis using trypanocides or insecticide-treated livestock [148].

In conclusion, the role of climate variability on the transmission of schistosomiasis in SSA cannot be over-emphasized. The impact of climate change on *Schistosoma* infection is greatly pronounced on the production, survivability, and fecundity rate of both freshwater snails and schistosomes. Therefore, all hands must be on deck in order to develop models capable of predicting the population dynamics of the *Schistosoma* species and snail host, as well as mapping the areas or regions in which climate change will have a profound impact on schistosomiasis transmission in sub-Saharan Africa. More so, there is a need for control measures for the disease in sub-Saharan Africa such as infrastructural development that can mitigate the effects of floods and development of vaccines and new drug formulations that will be capable of blocking re-infection of schistosomiasis and effectively overcome drug resistance. Since multidrug resistance to the worm has been reported [11,149] due to: the parasitic load; poor treatment compliance; co-infections of different strains of the parasites; and the rate of mutation of the parasite, as well as the ineffectiveness of praziquantel against the juvenile schistosomes [150]. Moreover, other measures that need to be put in place in order to lessen the burden of schistosomiasis in SSA include: an improvement and consistency in climate surveillance systems, execution of vaccination campaigns in target regions once vaccines are developed, sensitization of the public about the disease and its impacts on public health and public health policies, as well as improvement in support towards schistosomiasis research in order to better understand the current and possible future distribution of the disease.

## Figures and Tables

**Figure 1 ijerph-17-00181-f001:**
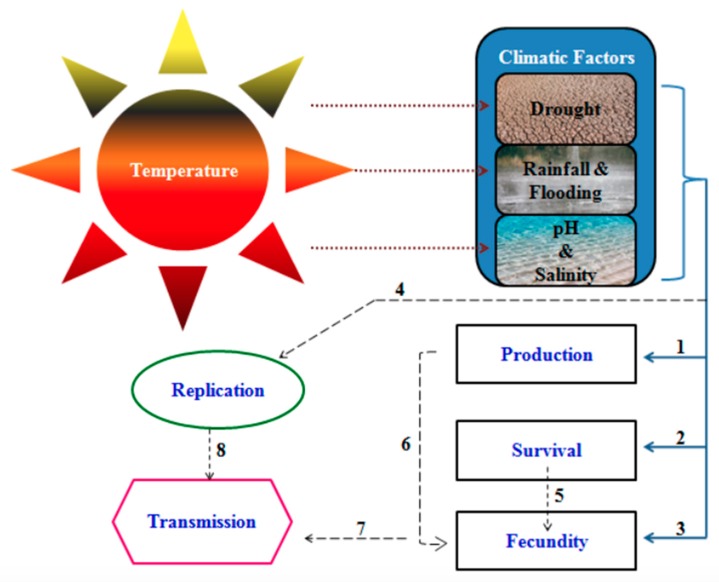
Proposed model eliciting effects of change in climatic factors on Schistosomiasis transmission. 1. Changes in climatic factors are an essential determinant in the production and development of snails, as well as the production and development time of miracidia and cercaria in the intermediate host (snails) and in freshwater depends on climate variability, 2. The snails. as well as miracidia and cercariae in freshwater and intermediate hosts. depends on a change in climatic factors to survive, 3. Climate changes pose a serious influence on the fecundity of the snails and hatching of schistosomes eggs into miracidia, 4. Replication of miracidia in the intermediate host to form sporocyte which in turn produce cercaria, 5. Increase in survival rate leads to an increase in fecundity rate, 6. Climate change enhances the rapid reproduction of snails and the rapid metamorphosis of miracidia into cercaria in the intermediate host, 7. Increase in the fecundity rate of snails enhances the transmission of schistosomiasis by increasing the production number of cercaria within snails, 8. The cercaria penetrates into the definitive human host, thereby metamorphosing into schistosomula that develop into adult worms which lay eggs that are responsible for the morbidity and mortality resulting from this disease.

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
