# Peer review of "The Effect of Climate Change and the Snail-Schistosome Cycle in Transmission and Bio-Control of Schistosomiasis in Sub-Saharan Africa"

_ijerph, 2019, doi:10.3390/ijerph17010181_

Round 1
Reviewer 1 Report
I believe the authors have adequately addressed the comments raised in the previous round of review. At this stage, I have no further substantial points to make. On a minor note, I suggest that the newly added paragraphs on the role of math modeling in epidemiology be carefully proofread, because in their present form they contain a number of typos that need to be fixed. A (possibly incomplete) list of suggested edits follows:
- l.540: model -> models
- l.542: model -> modeling
- l.543: the maturation
- l.548: although -> however
- l.550: unnecessary comma after "et al."
- l.552: biological -> biologically
- l.552: unnecessary semicolon after snails?
- l.553-4: "as predictions... environmental drivers" unclear meaning, please rephrase
- l.556: "Bi. Pfeifferi" should be italicized, and "pfeifferi" should not be capitalized
- l.556: which -> whose
- l.557: 2055 seems like a very specific point in time -- maybe something more nuanced?
- l.558-9: "McCreesh... Eastern Africa" makes it sound like the authors proposed a climate model that depends on snail ecology, please rephrase
- l.560: ... projections for the region. The sensitivity ...
- l.561: was -> were
- l.562: "S. mansoni" should be italicized
- l.563: "S. mansoni" should be italicized
- l.564: ... where control programmes are underway ... (or something like that)
- l.565: co-worker -> co-workers (or perhaps co-authors)
- l.567: ... in the laboratory. The model ...
- l.569: "in the rates of snails and cercariae" unclear, please rephrase
Author Response
Response to reviewer is attached herein

Reviewer 2 Report
This manuscript addresses the important public health goal of predicting the potential effects/impacts that climate change will have on the incidence and severity of human and animal shistosomiasis in Sub-Saharan Region in Africa. From a comprehensive literature review (150 papers), the authors hypothesize that climate change warming acompanied by increased rain fall will result in: 1) increased populations of infected vector snail populations in nearby village streams and lakes, and 2) the development and release of increased numbers of infective schistosome larval stages from the infected snails, which will greatly increase the risk of human infection in endemic areas. The ultimate goal of the project is the development of a model that can be used to monitor the status of transmission and determine the potential for increased transmission that may result from climate change in Sub-Saharan Region of Africa. Unfortunately the manuscript does not provide sufficient details on how this new model will function and how it will used to predict the impact of the schistosomaisis in the Region.
Author Response
Response to reviewer is attached herein.
Generally, the manuscript has been improved greatly.
We want to appreciate the Editor and the reviewer of our manuscript.

Reviewer 3 Report
Overall there is a certain amount of repetition which can be reduced, and in some areas a lack of numbers or data to 'prove' points made. For instance, a lot of temperature has a great effect on snail fecundity - but without stating what kind of effect and with low or high temp?
Needs to be checked for English, some issues with sentence structure, although generally not too egregious.
Abstract
Line 30-33: this sentence needs revision, a bit confusing as it is
Line 40: remove comma after decades
Line 43: have not hold
Line 46: Advisory group for NTDs
Line 46: remove these and namely, use a comma not a semicolon
Line 48: The instead of other
Line 54: also instead of otherwise
Line 57: Reference examples of ‘dire consequences’
Line 58: ‘In Africa there are four main species, S. mansoni ….’. Can use Abbreviation of Schistosoma as you already state the genus
Line 63: These references are for intercalatum, move them to a more appropriate sentence, and reference mekongi/japonicum papers here.
Line 65: Would reference a life cycle here – CDC or another paper which has a diagram
Line 68: reference disease states
Line 101: infected?
Line 101-104: Confused by this the three weeks of rainfall in the second sentence is different to the three weeks of rainfall in the first sentence? The rainy season is only three weeks?
Line 112: Increment by year?
Line 122: falciparum should be in italics as it is a species, also add the P.
Line 133: Snail species involved in transmission of Schistosomaisis – or something. Feels like missing the word host or needs rearranging.
Line 144: If it’s not schisto it doesn’t really matter for the purposes of this paper. Plus ref only mentions Egyptian snails, assume there will be more elsewhere in the world and even in Africa.
Should also mention in this paragraph which snails are in Africa.
Line 155: Schistosoma in italics as it is the genus name (ditto line 158), and reference
Line 160: ref
Line 162: Significantly different to what and how?
Line 172: But what are the roles? Surely role is very similar between species, so is environmental differences? Are some species living in drier areas, or wetter areas or areas with high…..sulfur in the water? What about genetic differences in the parasites infecting the different snail species? Are there subspecies?
Line 177: ponds
Line 184: Where is the ‘natural home’ of these species? How did the spread, through climate change or importation (accidental or directed)?
Line 189: to undergo asexual reproduction.
Line 191: Sooo there’s S. mansoni everywhere there is Biomphalaria snails?Do you mean distribution of S. mansoni? Is being infected healthy for the snails? Does it affect their reproduction?
Line 193: As neither of these snails occur in Africa, should this be included in this paper? I’m not sure how climate variability, flooding etc would impact on Oncomelania and Tricula snails as they are not present there. Would consider having as a short paragraph stating that these snails are important for schistosomiasis elsewhere, but are not present in Africa, and add it to the end of the second paragraph at the beginning of the section (3. Snail as an intermediate in the transmission of Schistosomaisis)
Line 228: Drive which way? Is it negative or positive association?
Line 231: remove comma
Line 234: an increase
Line 233-235: Is there an upper limit to this? Presumably there is a temperature range which the snails can live with and above and below that range they die.
Line 237: There must be a mistake in the previous sentence. Previous sentence states that increase temp decreases snail mortality – i.e. actually an increase temperature is good for snail survivability. In this sentence it states decrease snail production due to increase temps, so the snails die at higher temps. What would be more illuminating would be to state the temperature range of snail survivability (that has been tested anyway) here.
Line 238: So temperature dependent how, hotter they move and produce faster? Or hotter and they are less able to penetrate a snail host?
Line 240: The snail body temp? And again, how does it affect cercarial production.
Line 243: Ditto above comments. Temperature affects blah blah blah – but how? Negatively or positively? Increase or decrease parasite survivability?
Line 238: Ok, now were are at this paragraph which actually gives the details lacking in the previous two paragraphs, what is the point of the two previous paragraphs? They don’t provide meaningful data.
Line 251: Not sure what this means. Prior sentence indicates 15-31 snail pop is constant, so why are they decreasing when temp increased 14-26?
Line 257: Due to release of cercariae from snails presumably, any info on this?
Line 271: Correlated how….positive, negative…
Line 276: Here and in other places could cut ‘an co-workers’ down to et al
Line 281: Does turbulent flow negatively affect the snails?
Line 290: What do you mean by increase in population dynamics?
Line 308: Remove some – unless there were other models that showed the opposite…
Line 325: Mention that oncomelania is not in Africa.
Line 330-331: Delete of infections
Line 342: State when the drought was (year-year)
Line 348: They were tested and the result is not known, or that’s how many tested positive for schisto?
Line 377: ‘…salinity and time, it can or something.
Line 377: Bit confusing, pH on its own doesn’t affect survival but they survive at pH 8.2? So pH does matter?
Line 385: In this sense what is meant by conductivity? Higher salt content? So what is the relevant pH/salinity etc in Albert and Victoria that mean there is better survivability of the snails?
Line 422/424 % sings have an extra o
Line 484: italics B. africanus
Section 5 seems like a rehash of the previous sections (Paragraphs 3-6 in particular)– talks about temperature and salinity all over again. May be better to examine trends in temp/rain/salinity in the geographic areas – rather than the effect on the host, as that should have already been covered. Would consider putting any new info from this section into the sub-sections on temperature, rain, salinity etc Anything repetitive from the earlier sections, delete.
Line 540: models
Line 556: italics Bi. Pfeifferi
Line 562/563: italics S. mansoni
Line 565: contrast
Line 570: snails
Line 576/579/589/590: italics schistosoma
Line 592: no italics species
Control strategies for Schistosomiasis control – too many controls, delete second oneLine 600: Would say vegetation reduction instead of ‘association of snails with plants’ – which is more buzzword, less specific
Line 608: delete to
Line 616: Would definitely reference resistance to PZQ because as far as I am aware this has not been definitively shown, and may be down to under dosing which is particularly common in MDAs. Drug resistance has been shown primarily in lab studies (in vivo and in vitro), with low definitive drug resistance identified in the field.
Line 619: reinfection
Probably want to mention MDAs in this paragraph
Line 621-639: Best to stick to schisto. There is modelling on schisto done in China that you can look at. Ok to mention the malaria modelling, but briefly. As there are different mechanisms in malaria transmission and treatment (not least huge drug and insecticide resistance). As well as Zhang and Liang, can look for Gray and Williams – they have also done modelling in China. Is there any treatment modelling done in Africa? Or in South America at a stretch – the Asian modelling will also include animal hosts which are not as applicable in African schistosomiasis.
Line 672: italics Schistosoma
Author Response

(The authors gave the same response as above.)

Reviewer 4 Report
The paper is interesting and a valuable contribution to the field. However it is far too long and would be significantly enhanced if it were edited down by a third or even half. There is too much detail in areas that are only superficially relevant to the paper's main aims. These aims need to be clearly defined at the start of the paper, and then the aims achieved. This means deciding whether the paper is a presentation of evidence, a review (in which case a systematic approach would enhance this) or a modelling paper.
Author Response

(The authors gave the same response as above.)

Round 2
Reviewer 2 Report
1) In this revised manuscript the authors have responded to Reviewer comments by focusing the manuscript on the importance of developing mathematical models that can be used to predict the effects that climate change with warming and increased rainfall will have on increasing vector snail populations and the risk of transmission of schistosomiasis to human in sub-Saraharian Africa.
2) The critical information optained from models will be vital to the development of timely and effective strategies for controling vector snail populations and the preparations for treating cases of human schistosomiasis in the Region.
Author Response
Generally, the manuscript has been improved greatly.
We want to appreciate the Editor and the reviewer of our manuscript.
Reviewer 3 Report
Line 64: I don’t have access to either article ref’d so will have to take it on trust one of them refers to the life cycle.
This is quite picky, but for degrees can you please chage to the more usual degree sign °C rather than ◦C
Line 173: I am still confused about the ‘various roles’ snails have in transmission – various would indicate more than one role, but their role is to be infected, what are the other roles? Do you mean changes in susceptibility to schistosoma infection/geographic isolates in different areas?
Line 198: The updated sentence is confusing ‘number of intestinal schistosomiasis in regions where there is present of S. mansoni’ – prevalence? Intestinal schisto must refer to S. mansoni so why mention again? And didn’t really address my initial comment of how they are related – Biomphalaria must exist in places where schisto doesn’t exit, but schisto can’t exist in areas where Biomphalaria does not exist. So, is it rather that the distribution of S. mansoni in related to the presence of Biomphalaria snails.
Line 218: ‘optimal temperature (26 °C - 31 °C) can eventually..’
Line 225: “that is, temperature can either have positive or negative effects on the transmission of the 226 schistosomes from the intermediate host to the definitive host. The production of cercariae within the intermediate host is assisted by [higher???] temperatures (of what degrees), and this does not only help the production of cercariae but also plays a significant role in increasing the metabolic activity, energy and vitality of the snail to intensify the rate of cercarial production within the snail
Line 237: Still confused “Added to this, it was observed that the time needed for the snail population to decrease with a simultaneous increase in temperature ranges from 46 – 176 days, at a temperature between 14 – 26 °C.” Should the decrease be increase? If they have good survivability 15-31°C, why are the snails pops decreasing 14-26°C? Or is it conflicting information from two research papers, in which case you could say, ‘Conversely Blah et al observed etc etc’
Line 268: “alteration in the survival rate of cercariae” – decreased survivability?
|
40 |
Line 281 |
Turbulent affect the transmission of the schistosomiasis by altering the survival rate of cercariae |
But how
Line 364: “associations with salinity and time.” What were the associations? Salinity over time decreases snail pop/fecundity?
Line 507: I think delete most of this as you don’t really say much, plus it is in China, best to stick with the African example in the next para. “Mathematical modelling has been used in the Peoples Republic of China (PRC) to determine the effect of multi component integrated approaches in eliminating schistosomiasis there [ref]. Meanwhile, in Africa etc etc
Line 521: ‘simulated’
Line 637: No human vaccine available so perhaps state something about continued research into human schistosome vaccines
Author Response
Response to Reviewer 3: IJERPH-648037
As suggested by the first reviewer of our submitted manuscript IJERPH-648037, the following changes were effected:
Line 64: I don’t have access to either article ref’d so will have to take it on trust one of them refers to the life cycle.
Response: Centers for Disease Control and Prevention report on parasites-schistosomiasis, Reviews April 11, 2018 https://www.cdc.gov/parasites/schistosomiasis/gen_info/faqs.html
This is quite picky, but for degrees can you please chage to the more usual degree sign °C rather than ◦C
Response: The used of degree has been revised throughout the manuscript
Line 173: I am still confused about the ‘various roles’ snails have in transmission – various would indicate more than one role, but their role is to be infected, what are the other roles? Do you mean changes in susceptibility to schistosoma infection/geographic isolates in different areas?
Response: The word “various” has been deleted
Line 198: The updated sentence is confusing ‘number of intestinal schistosomiasis in regions where there is present of S. mansoni’ – prevalence? Intestinal schisto must refer to S. mansoni so why mention again? And didn’t really address my initial comment of how they are related – Biomphalaria must exist in places where schisto doesn’t exit, but schisto can’t exist in areas where Biomphalaria does not exist. So, is it rather that the distribution of S. mansoni in related to the presence of Biomphalaria snails.
Response: Was rephrased to “In sub-Saharan Africa, studies have shown that the wide geographical distribution of S. mansoni is related to the presence of Biomphalaria snail species in the region”
Line 218: ‘optimal temperature (26 °C - 31 °C) can eventually..’
Response: The change was effected.
Line 225: “that is, temperature can either have positive or negative effects on the transmission of the 226 schistosomes from the intermediate host to the definitive host. The production of cercariae within the intermediate host is assisted by [higher???] temperatures (of what degrees), and this does not only help the production of cercariae but also plays a significant role in increasing the metabolic activity, energy and vitality of the snail to intensify the rate of cercarial production within the snail
Response: “that is, temperature can either have positive or negative effects on the transmission of the schistosomes from the intermediate host to the definitive host” was deleted
“…… higher temperature of about 15 0C to 31 0C……” was rephrased
Line 237: Still confused “Added to this, it was observed that the time needed for the snail population to decrease with a simultaneous increase in temperature ranges from 46 – 176 days, at a temperature between 14 – 26 °C.” Should the decrease be increase? If they have good survivability 15-31°C, why are the snails pops decreasing 14-26°C? Or is it conflicting information from two research papers, in which case you could say, ‘Conversely Blah et al observed etc etc’
Response: “Added to this, it was observed that the time needed for the snail population to decrease with a simultaneous increase in temperature ranges from 46 – 176 days, at a temperature between 14 – 26 0C” was deleted
Line 268: “alteration in the survival rate of cercariae” – decreased survivability?
|
40 |
Line 281 |
Turbulent affect the transmission of the schistosomiasis by altering the survival rate of cercariae |
But how
Response: “An increase in water levels due to high rainfall may also cause water turbulence which may increase the flow rates of water that in turn disturb snail habitats as well as the decreased survivability of cercariae” was rephrased.
Line 364: “associations with salinity and time.” What were the associations? Salinity over time decreases snail pop/fecundity?
Response: “This was observed in a multivariate test which showed a significant decrease in the number of cercariae through the time period” was added.
Line 507: I think delete most of this as you don’t really say much, plus it is in China, best to stick with the African example in the next para. “Mathematical modelling has been used in the Peoples Republic of China (PRC) to determine the effect of multi component integrated approaches in eliminating schistosomiasis there [ref]. Meanwhile, in Africa etc etc
Response: “Studies [124,125] have predicted mathematical modelling has a multi-component integrated approach in eliminating schistosomiasis in People Republic of China. This has been employed in targeting the various transmission pathways and incorporating bovine vaccination (including SjCTPI-Hsp70), mass human and bovine chemotherapy and mollusciciding in the republic. In [126], Gray et al presented a baseline findings around the Dongting Lake in Hunan Province; the schistosomiasis endemic area in the People Republic of China. The results of the four-year multi-component intervention trial which comprises of human PZQ treatment, bovine vaccination and mollusciciding of Oncomelania snails, provided an insight into the feasibility of using a bovine transmission blocking vaccine as a component of integrated control. More so, the outcomes are potentially important implications for schistosomiasis elimination in people Republic of China and may possibly serve as a model for global control efforts”. Was deleted
“Mathematical modelling has been used in the people Republic of China (PRC) to determine the effect of multi component integrated approaches in eliminating schistosomiasis [124,125,126]. Meanwhile,….” Was added
Line 521: ‘simulated’
Response: “Simulation” was changed to “simulated”
Line 637: No human vaccine available so perhaps state something about continued research into human schistosome vaccines
Response “such as vaccination campaign once one of the human schistosome vaccine (BILHVAX, or the 28-kDa GST from S. haematobium) in trials is successful and approve” was rephrased.
Generally, the manuscript has been improved greatly.
We want to appreciate the Editor and the reviewer of our manuscript.
Reviewer 4 Report
I have reviewed the edited manuscript and am happy for it to be accepted in its current form, with minor typographical changes.
Author Response
Generally, the manuscript has been improved greatly.
We want to appreciate the Editor and the reviewer of our manuscript.
This manuscript is a resubmission of an earlier submission. The following is a list of the peer review reports and author responses from that submission.
Round 1
Reviewer 1 Report
The paper discusses the role that climate change may have on schistosomiasis transmission. The authors have brought together some interesting insights however the paper is a bit too general for the different snail systems being discusses, includes areas and need some work to improve it. I have attached a list of comments and changes to be made.

Author Response
Response to Reviewer 1: IJERPH-603058
As suggested by the first reviewer of our submitted manuscript IJERPH-603058, the following changes were effected
Page 2, line 59 to 60: the statement has been modified from “The disease has dire consequences on agricultural yields and grave effects on the life and development of school children and pregnant women in affected regions” to “The disease has dire consequences on child development, agricultural productivity and outcome of pregnancy in affected regions” Page 2, line 60 to 65: the statement has been modified from “The five causative agents identified as the main instigators of schistosomiasis are Schistosoma mansoni, Schistosoma japonicum, Schistosoma haematobium, Schistosoma mekongi and Schistosoma intercalatum. The first three are the most common and predominantly liable for hosting, transmitting and enhancing the occurrence of the infection in humans, while the last two are rare species that are restricted to a few central African countries” to “In Africa there are Schistosoma mansoni (intestinal), Schistosoma haematobium (urogenital), Schistosoma intercalatum (intestinal), Schistosoma guineensis (intestinal). The first two are common and wide spread while the last two are rare and restricted to central African countries. Schistosoma japonicum is predominant in China and a few other areas, but levels are low. Schistosoma mekongi is rare and associates with the mekong River” Page 2, line 70: the word “urinary” was changed throughout the manuscript to “urogenital” Page 4, line 156: the word “truncates” was changed throughout the manuscript to “truncates” Page 4, line 157: the statement “intermediate hosting was modified throughout the manuscript to “transmission” Page 4, line 157 to 159: “There are nine species of Schistosoma transmitted by Bulinus, three that infect human and six that infect Bovids or rodents, Bulinus can survive outside freshwater as they can aestivate” was added. Page 4, line 161: 50% of the total incidence was modified to “50% of the total incidence of Schistosoma” Page 4, line 163: “Among others” was added to the statement “This may be largely due to the wide geographical distribution of the Bulinus spp. (its intermediate host) in this region with the snail mostly endemic to Cameroon, Egypt and Senegal” Page 4, line 157: “The location of the parasite transmission consists of ponds and their prevalence depends on rainfall” was deleted Page 4, line 172: “makeup” was added to the statement “significant differences were observed in the genetic makeup of seven Bulinus spp” Page 4, line 180: the word freshwater or tropical pondwater was changed to “tropical freshwater pond” Page 4, line 181 to 182: “Several Biomphalaria species are known to survive within and outside freshwater habitats for a long time” was modified to “In natural setting, Biomphalaria species cannot survive outside the freshwater” Page 4, line 186: “glabrate” was changed throughout the manuscript to “glabrata” Page 5, line 194: “love” was changed to “suitable” Page 5, line 195 to 196: was rephrased to “There are many Biomphalaria species in places like Lake Victoria in Uganda where high transmission takes place” Page 5, line 196 to 198: was rephrased from “the wide geographical distribution of mansoni is closely related to the number of predisposed Biomphalaria snail species to the parasite, which assists in its asexual reproductive phase (sporocysts)” to “the wide geographical distribution of Biomphalaria snail species is closely related to the number of S. mansoni, because this snail species assists parasite in its asexual reproductive phase (sporocysts)” Page 5, line 204: “In the same vein” was changed to “more so” Page 5, line 205: “ megonki, S. sinensium (only in rodents)” was rephrased to “S. sinensium (only in rodents), S. megonki” Page 5, line 214: “prevalence” was changed to “presence” Page 5, line 217: “in some habitats” was added to the statement “Therefore, it is surmise to state that an increase in flooding activities in SSA in some habitats may result in severe health challenges including serious increases in schistosomiasis outbreaks” Page 5, line 227: “transmission” was added to the statement “Variations in the weather conditions have been recognized to have significant impact on the lifespan (mortality) and fecundity rate of both snails and worm’s transmission during the schistosome life-cycle” Page 5, line 233: “changes takes” was changed to “changes take” Page 5, line 238: “schistosomal” was changed to “schistosoma” Page 5, line 239: “As observed” was changed to “it has been shown” Page 6, line 242 to244: “Additionally, a rise in temperature may increase the infective stages of the parasite due to an increase in snail production and decrease in the growth and developmental rate of the parasite” was rephrased to “Additionally, an increase in temperature level may decrease the infective stages of the parasite due to decrease abundance in snail production and decrease in the growth and developmental rate of the parasite.” Page 6, line 247: “by an increase in temperature” was rephrased to “by body temperature” Page 6, line 254: “agent” was changed to “life cycle” Page 6, line 263 to 264: “according to the study, thus, this assertion needs further elucidation to be well understood” was added Page 6, line 267: “specie” was changed to “species” Page 6, line 272: “ mansoni worms” was changed to “S. mansoni cercariae” Page 6, line 289 to 290: was rephrased to “An increase in water levels due to high rainfall may also cause turbulent shear which may results in the alteration in the survival rate of cercariae through which the transmission of the disease is enhanced” Page 7, line 291: “Bulinus” was added to the statement “In years past, the considerable differences in the distribution of the Schistosoma parasite (Bulinus)” Page 7, line 323 “with reported cases of schistosomiasis such as Nigeria, South Africa, Niger, Senegal, Kenya, Malawi, Angola and Zimbabwe among others” was deleted Page 8, line 342: “Bulinus snail” was added to the statement “Studies have shown that during aestivation, uninfected Bulinus snail species have the…..” Page 7, line 368 to 369: “…..than 9 months can result in the eradication of schistosomiasis transmission in some…” was rephrased to “…..than 9 months can stop the transmission of schistosomiasis at that foci, while drought periods lasting less than….” Page 8, line 383: “However, it has been shown that cholerae has the ability to survive high-level pH of host cells [21]. Abnormal vaginal discharge also falls under infections or diseases caused by changes in pH, which in this case has effects on the vaginal flora” was deleted Page 8, line 397 to 405: the cercariae was identified molecularly Page 9, line 418: they are freshwater cercariae Page 9, line 417 to 419: “Unfortunately, no study has examined the effects of pH and conductivity on the production, survival and fecundity of Bulinus, Oncomelania and Tricula” was added Page 9, line 442: “Support for these findings comes from the study……: was rephrased to “This finding was supported by the study of Neto……” Page 13, line 579 to 583: was rephrased to “More so, there is need for control measures for the disease in sub-Saharan Africa such as infrastructural development that can mitigate the effects of floods, development of vaccines and new drug formulations that will be capable of blocking re-infection of schistosomiasis and abolish reported cases of drug resistance. Since multidrug resistance to the worm has been reported” Line 472 to 482: was rephrased in the manuscript (page 13, line 579 to 590) to “More so, there is need for control measures for the disease in sub-Saharan Africa such as infrastructural development that can mitigate the effects of floods, development of vaccines and new drug formulations that will be capable of blocking re-infection of schistosomiasis and abolish reported cases of drug resistance. Since multidrug resistance to the worm has been reported [141,142], due to parasitic load; poor treatment compliance; co-infections of different strains of the parasites; and the rate of mutation of the parasite, as well as the ineffectiveness of praziquantel against the juvenile schistosomes [143]. Moreover, other measures need to be put in place in order to lessen the burden of schistosomiasis in SSA includes: an improvement and consistency in climate surveillance systems, execution of vaccination campaigns in target regions once vaccines is developed, sensitization of the public about the disease and its impacts on public health and public health policies, as well as improvement in support towards schistosomiasis research in order to better understand the current and possible future distribution of the disease”
Generally, the manuscript has been improved greatly.
We want to appreciate the Editor and the reviewer of our manuscript.
Reviewer 2 Report
The manuscript by Dr. Adekiya and coauthors deals with a review of the effects of climatological factors and climate change on the environmental phases of the transmission cycle of schistosomiasis, a snail-transmitted flatworm infection that is endemic to many tropical regions of the developing world, most notably in Sub-Saharan Africa. The subject of this review is interesting and timely, as schistosomiasis still represents one of the most common parasitic diseases worldwide. The review appears to be quite comprehensive in the list of sub-topics that it covers, especially as far as the impacts of environmental factors and climate change on the population dynamics of the snail hosts are concerned. However, given the breadth of the abstract, I find that some potentially interesting ideas that are hinted to there turn our to be quite under-explored in the manuscript. Finally, the review is reasonably well written, although some effort should still be devoted to polish the writing style. A list of detailed comments follows.
Major comments
My major concern with this manuscript is that virtually no effort is devoted to reviewing existing modeling approaches aimed to describe the population dynamics of the snail species involved in the transmission of schistosomiasis, and/or the whole parasite transmission cycle. Of course, there could be material here for another full review (which I am not asking for), therefore clearly defining the boundaries of this exercise seems crucial. For instance, one could discuss existing models in terms of their ability to incorporate climate-related variables, such as temperature and/or rainfall. I believe that this is something fair to ask, as one of the objectives of this manuscript is to suggest "the need to develop an efficient and effective model which will predict Schistosoma spp. population dynamics". If this is one of the focal themes of the review, then a discussion of what can already be found in the literature seems to be warranted.
On a related note, some other points that are made in the abstract seem to be under-explored (if not touched upon at all) in the manuscript. This is the case, for instance, of control strategies (which are only briefly discussed in the concluding section of the review) or bio-control (which the authors seem to refer to only in the context of the development of new drugs). Discussing the role of climate change in these areas seems quite interesting an endeavour, therefore I was quite disappointed at finding almost nothing about these points in the actual review.
Finally, I believe that for some drivers, like rainfall, it would be interesting to elaborate on the role played by infrastructural interventions that are often put in place (also, and crucially for this review, in schistosomiasis-affected regions) to mitigate the effects of e.g. floods or droughts, such as dams and reservoirs. The presence of hydraulic infrastructure has often been correlated with schistosomiasis transmission, and I believe that this link could be usefully explored in the context of adaptations to climate change.
Minor comments
- l.22: unnecessary comma after extent
- l.24: have -> has
- l.25: which is further... -- unclear, what is exacerbated?
- l.47: 19 diseases... -- categorized by whom? NTD lists are institution-dependent, please specify
- l.54: The disease has dire... -- OK, but references are needed here
- l.59: for hosting, transmitting and enhancing... -- how can a causative agent host an infection?
- l.98: I'd replace evolution with dynamics (or something else) to avoid any possible misunderstanding
- l.122: population dynamics can't increase, do you mean population size?
- l.133: I'd say `the asexual phase of the reproduction cycle of the parasite'
- l.179: missing comma after Egypt
- l.185: is pleasant the best word choice here?
- l.193: comprises twenty
- l.213: it is necessary
- l.230: As observed... it has been shown... -- awkward construction
- l.233: increase the abundance?
- l.265: unnecessary comma after although
- l.266: it is believed that an increase and decrease -- almost tautological
- l.324: futuristic or future?
- l.374: cholerae
- l.385: including one
- Figure 1: is there any rationale behind the choice of different line styles, box shapes, etc? If so, please specify. If not, please consider uniforming. Also, the caption of the figure needs some rewriting, as it contains some very long sentences which make it difficult to digest
- l.440: cercariae
- l.441: intermediate hosts
- l.476: need to be
- l.636: I noted that refs 57 and 71. I don't know whether this is the only instance, but please double check the whole reference section
Author Response
Response to Reviewer 2: IJERPH-603058
As suggested by the first reviewer of our submitted manuscript IJERPH-603058, the following changes were effected
Major comments: two subsection has been introduces into the manuscript, and the concluding section of the manuscript has been modified. These sections reported the existing modelling approaches in describing the population dynamics of both the schistosomes and snail species involved in the transmission of schistosomiasis. More so, the section explained several control strategies which have been used over the years and predicted some other possible strategies that can be put in place.
Role of mathematical modelling in disease epidemiological studies
Mathematical modelling is a system used in describing real life problems in both mathematics language and concepts. The use of mathematical modelling comprises of several methods which include statistical models, dynamical systems, theoretical models or differential equations [105]. The applications of modelling have been proved very useful in natural sciences such as biology, meteorology, and earth sciences among others. Additionally, its role in the field of engineering, social science, medicine, physical system control and risk management cannot be overemphasized [106-109]. Also, mathematical modelling plays important role in the prevalence of infectious diseases, where it is useful in investigating or examining and quantifying the effect and cause of spread of infectious diseases [110,111]. Modelling can also be helpful in decision-making due to projected results generated such as changes in the pattern of disease spreads due to interventions [112].
Abiodun and colleagues [113] employed a mathematical modelling approach to study and examine the influence of temperature and rainfall on the population dynamics of Anopheles arabiensis. The results of their study precisely quantified the impacts of seasonal changes on the population dynamics of A. arabiensis and the vector (mosquito) over the study area. They further averred that their model was developed to efficiently predict the population dynamics of A. Arabiensis in other to assess various strategies that can be effective in the control of malaria [113]. More so, studies have been carried out on the outbreak of cholera, control strategy and population dynamics, co-infection of cholera with malaria using mathematical models. These studies were carried out to have the full understanding of the effect, causes of cholera and co-infection with other diseases such as malaria. They also highlight how cholera and co-infections with other diseases can be managed and prevent future occurrence [114-116]. Several other studies [117-119] have employed modelling technique in identification of causes and transmission dynamics of diseases such as tuberculosis (TB) and HIV.
Mangal and colleagues [120] modelled the effect of temperature on the worm burden and prevalence of schistosomiasis for optimal disease control strategy. It was observed that the burden of Schistosoma reached the climax at a temperature of 30 0C and drastically reduces when the temperature is raised to 35 0C. Therefore, it was concluded that the best stable temperature for the spread of schistosomiasis ranges between 20 0C to 35 0C, and that the best optimum temperature for the survival of Schistosoma parasite is 20 0C, which is the temperature that the parasite can survive at, and this can be helpful in the disease control. In another related study by McCreesh and Booth [62], the temperature-sensitive stage of S. mansoni and the life cycle of its Biomphilaria pfeifferi intermediate host were simulated. It was observed that the infection of S. mansoni in rivers and lakes was very high ranging between 15 – 19 0C and 20 – 25 0C respectively meanwhile, the survivability of snail reduces drastically outside the temperature range of 14 0C – 26 0C. In like manner, an epidemiological model was developed by Ngarakana-Gwasira and colleagues [26] to improve the prediction of the influence of climatic factors on the population dynamics and disparity of schistosomiasis strength in Zimbabwe. The study observed that the best temperature for the transmission of schistosomiasis in that region ranges between 18 0C to 28 0C and the optimal temperature for schistosoma transmission in this region was about 23 0C. Additionally, it was observed that the schistosoma infection in snails increase at 22 0C when compared to other temperature like 20 0C and 25 0C, and that the schistosoma parasite died when the temperature was raised to 30 0C [26]. Recently, mathematical modelling of temperature and rainfall influence on Schistosoma species population dynamics in South Africa was carried out by adopting schistosomiasis sub-model which incorporated climate parameters (temperature and rainfall). The study was employed to examine the impact of climate variability on the transmission dynamics of schistosomiasis. It was observed in the study that climate change poses great influence on the reproduction number of both schistosomes and freshwater snails[1].
Control strategies for Schistosomiasis control
Several preventive and control measures that have been employed over the years in eradicating and lessening the burden of this disease include the control of freshwater snail via: environmental control which comprises of temperature, current speed of water, water chemistry, light and shade, as well as association of snails with plants [121-124]. Other preventive control measures via intermediate snail host include; the use of molluscicides which can either be chemical (bayluscide, derivative of N-p-substituted phenyl uracil-5-sulphonamide, anilofos, fenitrothion, eugenol and thymol) or plants (Solanum nigrum, Ambrosia maritime, Thymelaea hirsute, Peganum harmala and Callistemon lauceolatus) as toxic agents against freshwater snail and the schistosomes [125, 126]. In addition, biological control which involve the use of different organisms such as Trematocranusplacodon fishes (e.g tilapia fish) and Cairinamaschata (muschovy ducks) as predators of freshwater snail [127]. Host parasite relationship is another control measure used in controlling the burden of this disease by making the freshwater snails to develop resistance to some schistosomes infection via the activation of cytokines such as TNF-α and IL-1 [128]. Other internal defence protein response in molluscan control is upregulation of granularin; a protein with phagocytotic activity against foreign particles. Increase in the dose or amount of schistosome in the freshwater is another means of controlling the intermediate snail host [129]. Additionally, genetic control is another means of controlling schistosomiasis intermediate snail host, by changing the strain of highly susceptible snails to non-susceptible ones and releasing the snails that are resistance to schistosome parasite into natural habitats such as, freshwater [130]. In the human host, praziquantel remain the only effective control in treating schistosomiasis, and it has been shown over the years that schistosomes have developed resistance against praziquantel as well as been unable to kill the immature worms in the human host. Unfortunately, all these preventive and control measures are time consuming and requires jumbo amount of money to be carried out, more so, these control measure fails to carter for total eradication and re-infection of schistosomiasis. Hence, the needs for an alternative control strategy.
Over the years, several studies have employed mathematical modelling methods in identifying the best control and cost-effective strategies in treatment and management of infectious diseases. The impact of chemotherapy on optimal control of malaria dynamics with infective treatments, introduction of infected immigrants and the use of insecticides against population dynamics of mosquitoes was described in a mathematical model carried out by Makinde and Okosun [131]. They observed immigrants who are infected with malaria has no considerable influence in the transmission of malaria, provided there is an existence of an effective treatment and mosquito control. They further averred that, the combination of treatment of infectives, assessment of infected immigrants and the use of insecticides in killing mosquitoes results in a better and effective control in the transmission of malaria. Another related study where an optimal control and cost-effectiveness analysis of three malaria preventive measures (spraying of mosquitos with insecticides, use of treated bednets and possible treatment of infected individual) was investigated by Okosun and co-workers [132]. It was discovered that the total cost for applying treated bednets and treatment of an infected individuals is not effective in eliminating malaria but sustainable. Also, it was shown that the combination of spraying mosquitoes with insecticides for 57 days at 100% and the use of 100% treatment for 20 days was very effective in eliminating malaria. Likewise, the use of the three controls (use of 100% treated bednets for 18 days, use of 100% insecticides spray and 87% use of treatment) can also be very effective in eliminating the disease though, it requires an unnecessary additional cost when compared to the strategy that uses insecticides spray and treatment of infected individuals. Interestingly and according to their model, they concluded that the combination of the three control strategies is the most cost-effective out of all the strategies employed in eliminating malaria [132].
Zhang and co-workers [133], developed a partial differential equations of schistosomiasis age-based system model in human hosts and its application in treatment strategies incorporating the human definitive host and intermediate snail host. The study suggested that the strategies of controlling the disease focusing on the rate of infection in age group maybe the most successful strategy in schistosomiasis treatment. However, they did not consider costs that will be effective in controlling a given age-dependent or the development of possible resistance of the parasite to chemotherapy due to mass drug administration [133]. Liang and co-workers [134], using a calibration approach and a mathematical model integrating various field data, designed as strategies control in the feasibility of characterizing site-specific schistosomiasis transmission in endemic villages of south-western Sichuan, China showed that a viable control strategy is important in lowering schistosomiasis transmission, and that the population dynamics can be reduced by focusing on snail control, chemotherapy and egg control [134]. Thus, it is therefore important to develop a time-dependent control strategies, in order to ascertain the best optimal strategy for schistosomiasis control, such as vaccination campaign and sensitization of the public about the disease, environmental control through the infrastructural interventions, improvement in the existing anti-schistosomal drugs and production of new drugs that can mitigate and abolish the re-infection of the disease, and also the control of snails species that are involved in the transmission of the disease.
The concluding part of the manuscript was modified
More so, there is need for control measures for the disease in sub-Saharan Africa such as infrastructural development that can mitigate the effects of floods, development of vaccines and new drug formulations that will be capable of blocking re-infection of schistosomiasis and abolish reported cases of drug resistance. Since multidrug resistance to the worm has been reported [141,142], due to parasitic load; poor treatment compliance; co-infections of different strains of the parasites; and the rate of mutation of the parasite, as well as the ineffectiveness of praziquantel against the juvenile schistosomes [143]. Moreover, other measures need to be put in place in order to lessen the burden of schistosomiasis in SSA includes: an improvement and consistency in climate surveillance systems, execution of vaccination campaigns in target regions once vaccines is developed, sensitization of the public about the disease and its impacts on public health and public health policies, as well as improvement in support towards schistosomiasis research in order to better understand the current and possible future distribution of the disease.
Minor comments
Page 1, line 23: the comma after extent was removed Page 1, line 25: have was changed to has Page 1, line 26: “which is further exacerbated partially by change in climate” was modified to “partially caused by change in climate” Page 2, line 47 to 55: the list of NTDs was provided according to WHO “According to 10th meeting of the World Health Organization (WHO) Strategic and Technical Advisory Group for Neglected Tropical Diseases in 2017, the number of NTDs has been increased to 20 following the addition of these 3 diseases namely; Scabies and other ectoparasites, snakebite envenoming, and chromoblastomycosis and other deep mycoses. Other existing 17 are buruli ulcer, chagas disease, dengue and Chikungunya, dracunculiasis (guinea worm disease), echinococcosis, foodborne trematodiases, human African trypanosomiasis (African sleeping sickness), leishmaniasis, leprosy (hansen's disease), lymphatic filariasis, onchocerciasis (river blindness), rabies, schistosomiasis, soil-transmitted helminths (STH) (ascaris, hookworm, and whipworm), trachoma, taeniasis/cysticercosis and yaws” Page 2, line 59: Reference was provided Page 2, line 59 to 64: was rephrased to “In Africa there are Schistosoma mansoni (intestinal), Schistosoma haematobium (urogenital), Schistosoma intercalatum (intestinal), Schistosoma guineensis (intestinal). The first two are common and wide spread while the last two are rare and restricted to central African countries. Schistosoma japonicum is predominant in China and a few other areas, but levels are low. Schistosoma mekongi is rare and associates with the mekong River [5,6]” Page 2, line 102: “evolution” was replaced with : “dynamics” Page 3, line 126: population dynamics was rephrased to “transmission dynamics and the population size” Page 3, line 137: “The asexual phase of the parasite takes…….” Was rephrased to “The asexual phase of the reproduction cycle of the…..” Page 4, line 185: comma was put at after Egypt. Page 5, line 191: “pleasant” was changed to “suitable” Page 5, line 200: “Tricula comprises of twenty….” Was rephrased to “There are twenty species of Tricula under the Triculinae subfamily…” Page 5, line 220: “Therefore, it necessary…..” was rephrased to “Therefore, it is necessary…….” Page 5, line 237: “As observed……” was changed to “It has been shown…..” Page 6, line 240 to 242: “Additionally, a rise in temperature may…….” Was rephrased to “Additionally, an increase in temperature level may decrease the infective stages of the parasite due to decrease abundance in snail production….” Page 6, line 274: comma was removed after although Page 6, line 275 to 276: “……that an increase and decrease…..” was rephrased to “……..that an increase or a decrease……” Page 7, line 332: “futuristic” was changed to “future” Page 8, line 381: “Vibrio cholera” was corrected to “Vibrio cholerae” Page 9, line 389: “includes one” was rephrased to “including one….” In the figure section, the rationale and choice of box shapes depict different parameters and the roles of climate change in transmission dynamics of schistosomiasis. More so, the captions of the figure was modified. Page 10, line 443: “cercaria” was changed to “cercariae” Page 10, line 443: “intermediate host” was changed to “intermediate hosts” The conclusion has been modified. Double references in the references section was corrected.Generally, the manuscript has been improved greatly.
We want to appreciate the Editor and the reviewer of our manuscript.
Reviewer 3 Report
1. This manuscript clearly and thoroughly describes the potential impact that global warming could have on increasing populations of Biomphalaria and Bulinus species, the vectors snails for intestinal and urinary schistosomiasis in the sub-Saharan Region.
2. In order to respond to increased outbreaks due to global warming, the investigators appropriately advocate for resources to evaluate the potential impact that bio-control measures could have for controlling schistosomiasis in sub-Saharan Africa, where re-infection in populations continues in many residents despite treatment with praziquantel.
3. Based on their findings on the impact of climate change on the incidence of schistosomiasis in the sub Saharan Region, the authors propose to develop efficient models capable of predicting which areas or regions will be at risk of schistosomiasis outbreaks because of global warming.
Author Response
Response to Reviewer 3: IJERPH-603058
Generally, the manuscript has been improved greatly.
We want to appreciate the Editor and the reviewer of our manuscript.
Reviewer 4 Report
This review paper concerns the impact of climate change on schistosomiasis transmission in sub-Saharan Africa. The major part of the paper is devoted to reviewing the disease cycle, the distribution of several intermediate hosts in sub-Saharan Africa and the effects of temperature, rainfall, flooding, drought, pH and conductivity on those hosts. While the Abstract suggests that a major objective of the paper is to argue for the development of mathematical models to assess “bio-control” mechanisms to address re-infection risks, there is no clear definition of bio-control or, more importantly a discussion of the type of mathematical models that would be useful. There is an extensive literature on the modeling of schistosomiasis, although mainly related to mass drug administration as the control intervention. There is indeed a need for models addressing the environmental determinants of transmission of schistosomiasis and reference is made in the paper to studies of other infectious diseases where this has been the objective. However, the nature of the models used and their potential application to the environmental determinants of schistosomiasis transmission are not addressed.
In short, as currently focused, the paper adds little to the recent papers specifically addressing schistosomiasis and climate change, for example, those of Yang and Berquist, Trop. Med. Inf. Dis, 2018; McCreesh et al, Parasites and Vectors, 2015 and Blum and Hotez, 2018. In addition, there is no mention of the social and economic factors that will also respond to climate change and simultaneously affect schistosomiasis transmission. These factors are often acknowledged, but seldom addressed, in the infectious disease literature. Development is proceeding quickly in some parts of Africa and more slowly in others. How that development map might relate to schistosomiasis prevalence in sub-Saharan Africa would be of considerable interest, both currently and projected into the future. It has been argued that development has been a major factor in China’s successes in schistosomiasis control, Spear et al, Trop Med Inf Dis 2017.
There are a variety of other issues in the present draft and I mention only a few for brevity. The term population dynamics is used in unusual ways and confusing ways. For example, at line 121 an “increase in population dynamics” is cited as depending on seasons. Not sure what is meant. Similarly, at line 294 and in a slightly different context at line 482. At line 61 it is implied the humans are the only definitive hosts. This is certainly not the case for S. japonicum where bovines are important hosts and various other mammals can be infected as well.
Author Response
Response to Reviewer 4: IJERPH-603058
As suggested by the first reviewer of our submitted manuscript IJERPH-603058, the following changes were effected
We have included the highlighted subsection into the manuscript, these section explained the several existing modelling approaches in describing the population dynamics of both the schistosomes and snail species involved in the transmission of schistosomiasis. More so, the section explained several control strategies which have been used over the years and predicted some other possible strategies that can be put in place
Role of mathematical modelling in disease epidemiological studies
Mathematical modelling is a system used in describing real life problems in both mathematics language and concepts. The use of mathematical modelling comprises of several methods which include statistical models, dynamical systems, theoretical models or differential equations [105]. The applications of modelling have been proved very useful in natural sciences such as biology, meteorology, and earth sciences among others. Additionally, its role in the field of engineering, social science, medicine, physical system control and risk management cannot be overemphasized [106-109]. Also, mathematical modelling plays important role in the prevalence of infectious diseases, where it is useful in investigating or examining and quantifying the effect and cause of spread of infectious diseases [110,111]. Modelling can also be helpful in decision-making due to projected results generated such as changes in the pattern of disease spreads due to interventions [112].
Abiodun and colleagues [113] employed a mathematical modelling approach to study and examine the influence of temperature and rainfall on the population dynamics of Anopheles arabiensis. The results of their study precisely quantified the impacts of seasonal changes on the population dynamics of A. arabiensis and the vector (mosquito) over the study area. They further averred that their model was developed to efficiently predict the population dynamics of A. Arabiensis in other to assess various strategies that can be effective in the control of malaria [113]. More so, studies have been carried out on the outbreak of cholera, control strategy and population dynamics, co-infection of cholera with malaria using mathematical models. These studies were carried out to have the full understanding of the effect, causes of cholera and co-infection with other diseases such as malaria. They also highlight how cholera and co-infections with other diseases can be managed and prevent future occurrence [114-116]. Several other studies [117-119] have employed modelling technique in identification of causes and transmission dynamics of diseases such as tuberculosis (TB) and HIV.
Mangal and colleagues [120] modelled the effect of temperature on the worm burden and prevalence of schistosomiasis for optimal disease control strategy. It was observed that the burden of Schistosoma reached the climax at a temperature of 30 0C and drastically reduces when the temperature is raised to 35 0C. Therefore, it was concluded that the best stable temperature for the spread of schistosomiasis ranges between 20 0C to 35 0C, and that the best optimum temperature for the survival of Schistosoma parasite is 20 0C, which is the temperature that the parasite can survive at, and this can be helpful in the disease control. In another related study by McCreesh and Booth [62], the temperature-sensitive stage of S. mansoni and the life cycle of its Biomphilaria pfeifferi intermediate host were simulated. It was observed that the infection of S. mansoni in rivers and lakes was very high ranging between 15 – 19 0C and 20 – 25 0C respectively meanwhile, the survivability of snail reduces drastically outside the temperature range of 14 0C – 26 0C. In like manner, an epidemiological model was developed by Ngarakana-Gwasira and colleagues [26] to improve the prediction of the influence of climatic factors on the population dynamics and disparity of schistosomiasis strength in Zimbabwe. The study observed that the best temperature for the transmission of schistosomiasis in that region ranges between 18 0C to 28 0C and the optimal temperature for schistosoma transmission in this region was about 23 0C. Additionally, it was observed that the schistosoma infection in snails increase at 22 0C when compared to other temperature like 20 0C and 25 0C, and that the schistosoma parasite died when the temperature was raised to 30 0C [26]. Recently, mathematical modelling of temperature and rainfall influence on Schistosoma species population dynamics in South Africa was carried out by adopting schistosomiasis sub-model which incorporated climate parameters (temperature and rainfall). The study was employed to examine the impact of climate variability on the transmission dynamics of schistosomiasis. It was observed in the study that climate change poses great influence on the reproduction number of both schistosomes and freshwater snails[1].
Control strategies for Schistosomiasis control
Several preventive and control measures that have been employed over the years in eradicating and lessening the burden of this disease include the control of freshwater snail via: environmental control which comprises of temperature, current speed of water, water chemistry, light and shade, as well as association of snails with plants [121-124]. Other preventive control measures via intermediate snail host include; the use of molluscicides which can either be chemical (bayluscide, derivative of N-p-substituted phenyl uracil-5-sulphonamide, anilofos, fenitrothion, eugenol and thymol) or plants (Solanum nigrum, Ambrosia maritime, Thymelaea hirsute, Peganum harmala and Callistemon lauceolatus) as toxic agents against freshwater snail and the schistosomes [125, 126]. In addition, biological control which involve the use of different organisms such as Trematocranusplacodon fishes (e.g tilapia fish) and Cairinamaschata (muschovy ducks) as predators of freshwater snail [127]. Host parasite relationship is another control measure used in controlling the burden of this disease by making the freshwater snails to develop resistance to some schistosomes infection via the activation of cytokines such as TNF-α and IL-1 [128]. Other internal defence protein response in molluscan control is upregulation of granularin; a protein with phagocytotic activity against foreign particles. Increase in the dose or amount of schistosome in the freshwater is another means of controlling the intermediate snail host [129]. Additionally, genetic control is another means of controlling schistosomiasis intermediate snail host, by changing the strain of highly susceptible snails to non-susceptible ones and releasing the snails that are resistance to schistosome parasite into natural habitats such as, freshwater [130]. In the human host, praziquantel remain the only effective control in treating schistosomiasis, and it has been shown over the years that schistosomes have developed resistance against praziquantel as well as been unable to kill the immature worms in the human host. Unfortunately, all these preventive and control measures are time consuming and requires jumbo amount of money to be carried out, more so, these control measure fails to carter for total eradication and re-infection of schistosomiasis. Hence, the needs for an alternative control strategy.
Over the years, several studies have employed mathematical modelling methods in identifying the best control and cost-effective strategies in treatment and management of infectious diseases. The impact of chemotherapy on optimal control of malaria dynamics with infective treatments, introduction of infected immigrants and the use of insecticides against population dynamics of mosquitoes was described in a mathematical model carried out by Makinde and Okosun [131]. They observed immigrants who are infected with malaria has no considerable influence in the transmission of malaria, provided there is an existence of an effective treatment and mosquito control. They further averred that, the combination of treatment of infectives, assessment of infected immigrants and the use of insecticides in killing mosquitoes results in a better and effective control in the transmission of malaria. Another related study where an optimal control and cost-effectiveness analysis of three malaria preventive measures (spraying of mosquitos with insecticides, use of treated bednets and possible treatment of infected individual) was investigated by Okosun and co-workers [132]. It was discovered that the total cost for applying treated bednets and treatment of an infected individuals is not effective in eliminating malaria but sustainable. Also, it was shown that the combination of spraying mosquitoes with insecticides for 57 days at 100% and the use of 100% treatment for 20 days was very effective in eliminating malaria. Likewise, the use of the three controls (use of 100% treated bednets for 18 days, use of 100% insecticides spray and 87% use of treatment) can also be very effective in eliminating the disease though, it requires an unnecessary additional cost when compared to the strategy that uses insecticides spray and treatment of infected individuals. Interestingly and according to their model, they concluded that the combination of the three control strategies is the most cost-effective out of all the strategies employed in eliminating malaria [132].
Zhang and co-workers [133], developed a partial differential equations of schistosomiasis age-based system model in human hosts and its application in treatment strategies incorporating the human definitive host and intermediate snail host. The study suggested that the strategies of controlling the disease focusing on the rate of infection in age group maybe the most successful strategy in schistosomiasis treatment. However, they did not consider costs that will be effective in controlling a given age-dependent or the development of possible resistance of the parasite to chemotherapy due to mass drug administration [133]. Liang and co-workers [134], using a calibration approach and a mathematical model integrating various field data, designed as strategies control in the feasibility of characterizing site-specific schistosomiasis transmission in endemic villages of south-western Sichuan, China showed that a viable control strategy is important in lowering schistosomiasis transmission, and that the population dynamics can be reduced by focusing on snail control, chemotherapy and egg control [134]. Thus, it is therefore important to develop a time-dependent control strategies, in order to ascertain the best optimal strategy for schistosomiasis control, such as vaccination campaign and sensitization of the public about the disease, environmental control through the infrastructural interventions, improvement in the existing anti-schistosomal drugs and production of new drugs that can mitigate and abolish the re-infection of the disease, and also the control of snails species that are involved in the transmission of the disease.
The concluding part of the manuscript was modified
More so, there is need for control measures for the disease in sub-Saharan Africa such as infrastructural development that can mitigate the effects of floods, development of vaccines and new drug formulations that will be capable of blocking re-infection of schistosomiasis and abolish reported cases of drug resistance. Since multidrug resistance to the worm has been reported [141,142], due to parasitic load; poor treatment compliance; co-infections of different strains of the parasites; and the rate of mutation of the parasite, as well as the ineffectiveness of praziquantel against the juvenile schistosomes [143]. Moreover, other measures need to be put in place in order to lessen the burden of schistosomiasis in SSA includes: an improvement and consistency in climate surveillance systems, execution of vaccination campaigns in target regions once vaccines is developed, sensitization of the public about the disease and its impacts on public health and public health policies, as well as improvement in support towards schistosomiasis research in order to better understand the current and possible future distribution of the disease.
Page 2, line 65 to 67: “The life cycles of all the Schistosoma spp are all similar yet very complex as the parasite alternates between two hosts: the intermediate (snail) and the definitive (human) host. Infection in the definitive….” Was modified to “The life cycles of all the Schistosoma spp are all similar yet very complex as the parasite alternates between two hosts: the intermediate (snail) and the definitive (such as human, bovines and domestic cattle) host. In the human hosts………” Page 3, line 126: population dynamics was rephrased to “transmission dynamics and the population size” Page 7, line 302: “……increase in the population dynamics of schistosomiasis…..” was rephrased to “………..increase in the transmission dynamics of schistosomiasis…..” Page 13, line 568: “……..population dynamics…….: was changed to “……transmission dynamics……”
Generally, the manuscript has been improved greatly.
We want to appreciate the Editor and the reviewer of our manuscript.
Round 2
Reviewer 2 Report
The authors have done a decent job of addressing the comments raised during the first round of review. In my view, though, some important issues still remain open:
- the title of the manuscript seems to suggest that not only transmission, but also (bio-)control of schistosomiasis is going to be discussed in the context of a changing climate ("Climate change ... as a predictor", whatever that "predictor" actually means). I do appreciate that the authors have addressed one of the points I have previously raised concerning the lack of detail on control strategies. However, there is no effort yet to discuss controls in the context of climate change. If the authors think that is something they are ready to add to the manuscript, that would be one of the highlight of their contribution; if not -- fine, but I would suggest at least to update the title of the manuscript;
- I also appreciate that the authors have included in their revision an overview of some of the modeling approaches that have been proposed to describe schistosomiasis transmission dynamics. However, this addition now begs the question: in the authors' opinion, what is missing from existing modeling approaches to "advocate a bid" to build new and better models? I think this point deserves some sort of discussion in the manuscript;
- relatedly, I am not totally convinced that it is indeed necessary to review modeling approaches for malaria or cholera (for which a shelf-filling literature exists, in particular for the former). I think limiting the review effort to schistosomiasis (as far as modeling in concerned, but possibly also for climate-related discussions) would improve the focus of this contribution;
- finally, additional care must be put in checking and proofreading the manuscript, because a number of typos, awkward sentences and technicalities in need of a fix are still present (see below for a relatively extensive, yet possibly incomplete list).
Minor comments
- l.33: remove "prediction of"
- l.34: remove "of the disease"
- l.46: According to the
- l.125: population dynamics cannot increase (or decrease, for that matter), here and everywhere else
- l.126: "increase in transmission dynamics" sounds awkward too
- l.142: cercariae, which
- l.143: parasite. Cercariae have
- l.212: "It is surmise" I cannot actually understand whether you actually agree or disagree with what follows. I would suggest to rewrite this sentence to improve its clarity
- l.213: "flooding activities" sounds off -- "flooding" alone may do it, here and everywhere
- l.217: impacts (or may impact, depending of the level of confidence you want to attribute to this sentence)
- l.235: "It has been shown" is repeated twice in the sentence
- l.238: "decrease the infective stages" is unclear
- l.250: use consistently abbreviations for binomial nomenclature that has already been used previously in the manuscript, here and everywhere else
- l.271: I repeat that this sentence has almost no information content, especially in places where the hydrological balance is heavily rainfall-driven
- l.286: "Schistosoma parasite (Bulinus)" what do you mean?
- l.297: "Similar to an increase in rainfall" where does this come from? As far as I know, climate projections concerning rainfall are quite spatially heterogeneous for SSA, i.e. possibly with northern/eastern regions experiencing an increase in rainfall and southern regions experiencing the opposite -- also depending on emission scenarios and the likes (see e.g. Serdeczny et al. 2016, Reg Environ Change). I would suggest an ounce of care in stating climate projections in a casual way, especially in a manuscript that is heavily focused on climate change and some of its effects on health
- l.299: "availability to limited resources" is an awkward expression
- l.327: remove comma after Oncomelania
- l.359: "Therefore, it is safe" is it, though? Ref. 10 is not exactly focused on droughts, therefore it is not straightforward to conclude (at least for me) whether "the increased risk of schistosomiasis transmission" is associated with the dry period, the beginning of the rainy season of any other factor
- l.408: is salinity a climatic factor?
- l.432: which are highly
- l.497: involves
- l.502: "increase in the amount of schistosomes in freshwater" I do not really understand what the authors mean here
- l.509: "fail to carter for" is unclear
- l.510: disease, hence the need
- l.515: malaria have
- l.521: how can a cost be (not) effective? Unclear
- l.538: either thus or therefore
- l.560: "population dynamics of schistosomiasis transmission" is an awkward expression
- l.563: species should not be italicized
- l.563: as well as of mapping
- l.567: "abolish reported cases" seems to suggest the reports of drug resistance should be purged from epidemiological records
- l.570: measures that need
- l.572: vaccines are
Reviewer 4 Report
The changes in the revised version do not go to my earlier criticisms other than to acknowledge that models have been used to demonstrate the potential of integrated control strategies that include environmental effects. The bulk of the paper remains focused on a review of the effect of temperature and other environmental variables on various snail species relevant to schistosomiasis transmission. However, there is no linkage of this review to particular geographical settings and their resident snail populations that might inform forecasts of climate change in any useful way.